# CARE: Confidence-Aware Ratio Estimation for Medical Biomarkers

**Jiameng Li**[1]                                            JIAMENG.LI@KULEUVEN.BE
**Teodora Popordanoska**[1]                    TEODORA.POPORDANOSKA@KULEUVEN.BE
**Aleksei Tiulpin**[2,3]                               ALT4026@MED.CORNELL.EDU
**Sebastian G. Gruber**[1]                       SEBASTIAN.GRUBER@KULEUVEN.BE
**Frederik Maes**[1]                                 FREDERIK.MAES@KULEUVEN.BE
**Matthew B. Blaschko**[1]                      MATTHEW.BLASCHKO@KULEUVEN.BE

[1] *KU Leuven*      [2] *University of Oulu*      [3] *Weill Cornell Medicine*

**Editors:** Accepted for publication at MIDL 2026

## Abstract

Ratio-based biomarkers (RBBs), such as the proportion of necrotic tissue within a tumor, are widely used in clinical practice to support diagnosis, prognosis, and treatment planning. These biomarkers are typically estimated from segmentation outputs by computing region-wise ratios. Despite the high-stakes nature of clinical decision making, existing methods provide only point estimates, offering no measure of uncertainty. In this work, we propose a unified *confidence-aware* framework for estimating ratio-based biomarkers. Our uncertainty analysis stems from two observations: (1) the probability ratio estimator inherently admits a statistical confidence interval regarding local randomness (bias and variance); (2) the segmentation network is not perfectly calibrated (calibration error). We perform a systematic analysis of error propagation in the segmentation-to-biomarker pipeline and identify model miscalibration as the dominant source of uncertainty. Extensive experiments show that our method produces statistically sound confidence intervals, with tunable confidence levels, enabling more trustworthy application of segmentation-derived RBBs in clinical workflows.
**Codes:** https://github.com/renaissanceee/care
**Keywords:** Medical Imaging Analysis, Uncertainty Quantification, Trustworthy AI

## 1. Introduction

The success of deep learning in medical image analysis, particularly since the introduction of UNet architectures (Ronneberger et al., 2015; Isensee et al., 2021), has enabled automated segmentation of anatomical and pathological structures across a range of clinical imaging tasks. However, segmentation is rarely the end goal in clinical practice. Instead, it often serves as an intermediate step towards quantifying tissue biomarkers, such as volumes (Popordanoska et al., 2021; Rousseau et al., 2025; Kazerouni et al., 2023; Abdusalomov et al., 2023) and fraction scores (Ronneberger et al., 2015; Isensee et al., 2021; Bahna et al., 2022; Kim et al., 2008; Solovyev et al., 2020) that are used to assess disease progression, guide treatment decisions, or monitor therapeutic responses. The ratio-based biomarkers are of specific interest in this paper, which are typically derived from two volume measurements computed from pixel-wise predictions. We note here that the naive computation of an RBB from a standard segmentation model does not offer uncertainty quantification (UQ), which

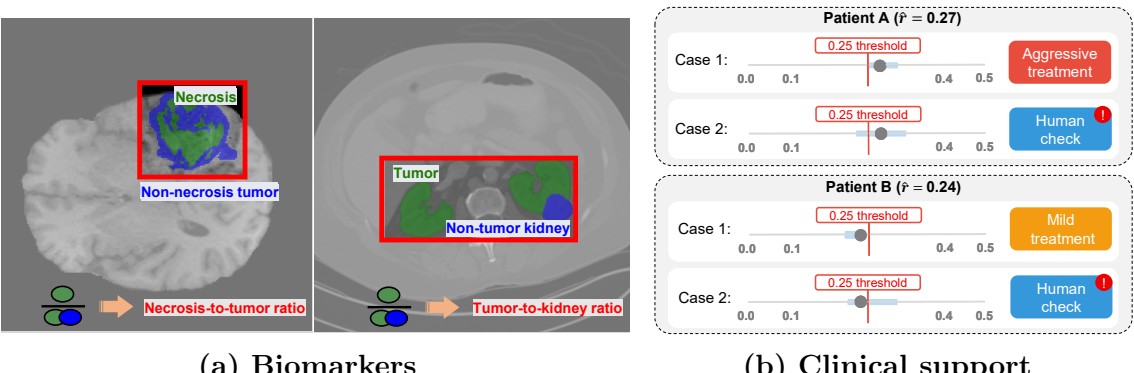

(a) **Biomarkers**    (b) **Clinical support**

Figure 1: Examples of ratio-based biomarkers and their roles in clinical support. (a): Ratio-based biomarkers (Baid et al., 2021; Myronenko et al., 2023) exist in many organs and modalities. (b): An illustrative example where a high-risk threshold is defined as 0.25; CARE calls for human check when confidence intervals cross the thresholds.

limits the clinical adoption and undermines its reference value for decision-making. To address this, we study confidence-aware ratio estimation for RBBs.

Fig. 1 (a) shows examples of two clinically used RBBs: necrosis-to-tumor ratio (NTR) and tumor-to-kidney ratio (TKR). The NTR is mostly used in brain cancer treatment (Henker et al., 2019, 2017) to quantify the proportion of necrotic (non-viable) tissue within a tumor. TKR (Herts et al., 2002) indicates the extent of tumor infiltration within the kidney and is computed (mainly) from abdominal CT. A straightforward method for computing these ratios involves using segmentation models to identify the subregion and the whole foreground region, and then calculating the ratio based on averaged softmax confidence scores over these regions. However, the interpretation of this point estimate can change once the confidence interval is considered. Following the example in Fig. 1 (b), consider a clinical threshold of 0.25 for initiating aggressive treatment. Based on point estimates alone, Patient A would receive aggressive treatment (high ratio) while Patient B would receive mild treatment (low ratio).However, if the associated confidence interval spans the decision threshold (case 2), the estimation is flagged for mandatory expert review to mitigate potential misdiagnosis risk. Such double-check procedures are essential in clinical practice, as they provide an additional safeguard for patients and enhance the robustness of downstream decision-making.

Despite the clinical importance of quantifying uncertainty, most efforts continue to focus on improving the accuracy of the upstream segmentation (Ronneberger et al., 2015; Isensee et al., 2021; Hatamizadeh et al., 2021). We propose CARE, a framework for estimation of confidence intervals in RBBs that is mathematically grounded, does not require additional training or sampling at test time. Our core contribution lies in the identification of sources of error and quantifying their individual impacts on the overall confidence intervals (Fig. 2).. Specifically, we establish a ratio estimator bound using Markov's inequality (Resnick, 2003) and derive a squared error estimator from volume predictions. To quantify the error caused by miscalibration, we provide theoretical insights into the relationship between model calibration and ratio estimation and propose a miscalibration-based bound, building on recent advances in calibration error (CE) estimation (Guo et al., 2017; Popordanoska et al., 2022) and a

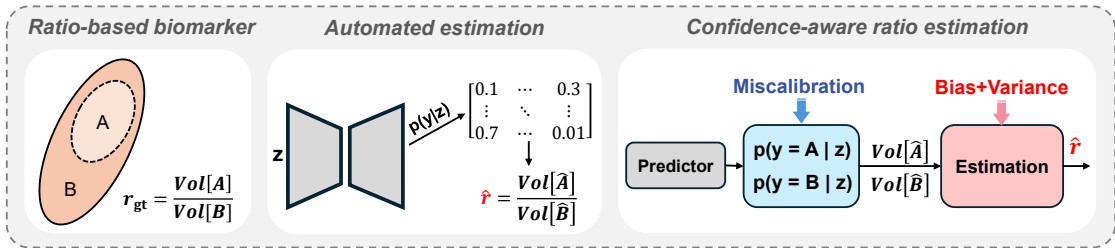

Figure 2: Overview. In automated medical imaging analysis, biomarkers are often computed from network predictions. To quantify the uncertainty of ratio-based biomarkers, we introduce CARE, a confidence-aware estimation method providing reliable confidence intervals.

recently established connection between volume estimation bias and CE. In summary, our main contributions are:

1. We propose CARE, a principled framework for trustworthy estimation of ratio-based biomarkers in an automated estimation workflow with minimum assumptions.

2. We analyze the sources of error across the entire segmentation-to-biomarker pipeline and empirically demonstrate that miscalibration is the dominant factor.

3. Experiments confirm that CARE effectively tracks the prediction uncertainty, evidenced by its coverage of erroneous predictions and its distinguishability of segmentation difficulties. In addition, CARE yields tighter confidence intervals than other sound adaptive uncertainty quantification methods.

## 2. Preliminaries

We define the RBB as a ratio between volumes $V_A$ and $V_B$ (Henker et al., 2019, 2017). The ratio is estimated via a standard segmentation framework, where $V_A$ and $V_B$ are calculated from predicted probabilities.

**Definition 1 (Ratio from Segmentation Networks)** *Given per-pixel inputs $\{z_i\}_{i=1}^n$, labels $\{y_{A,i}, y_{B,i}\}_{i=1}^n$ and segmentation model $g: z_i \to g_A(z_i), g_B(z_i) \in [0,1]$, the labeled ratio $r_{\mathrm{gt}}$ and predicted ratio $\hat{r}$ within $n$ pixels are calculated by:*

$$r_{\mathrm{gt}} = \frac{\bar{y_A}}{\bar{y_B}} = \frac{\sum_{i=1}^n y_{A,i}}{\sum_{i=1}^n y_{B,i}}, \ and \ \hat{r} = \frac{\bar{g_A}}{\bar{g_B}} = \frac{\sum_{i=1}^n g_A(z_i)}{\sum_{i=1}^n g_B(z_i)}. \tag{1}$$

**Definition 2 (Confidence Interval)** *Let $\hat{r}$ be an estimator of an unknown but fixed true value $r_{\mathrm{gt}}$, and let $\alpha \in (0,1)$ be a given significance level. The $(1-\alpha)$ confidence interval for $r_{\mathrm{gt}}$ is a interval $[\epsilon_l, \epsilon_u]$ such that*

$$\mathbb{P}(\epsilon_l \leq r_{\mathrm{gt}} \leq \epsilon_u) \geq 1 - \alpha. \tag{2}$$

**Definition 3 (Empirical Coverage Rate)** *Let $\mathcal{D}_{\text{test}} = \{(\hat{r}_i, r_{\text{gt},i})\}_{i=1}^n$ be a test set of n samples, where $\hat{r}_i, r_{\text{gt},i}$ are the prediction and the ground-truth. Suppose a UQ method provides a confidence interval $[\epsilon_{l,i}, \epsilon_{u,i}]$ for $\hat{r}_i$ at a confidence $(1 - \alpha)$. The empirical coverage rate measures the proportion of samples whose true values fall within the confidence interval:*

$$\text{Coverage}(\mathcal{D}_{\text{test}}) = \frac{1}{n} \sum_{i=1}^{n} \mathbf{1}(\epsilon_{l,i} \leq r_{\text{gt},i} \leq \epsilon_{u,i}), \tag{3}$$

*where $\mathbf{1}(\cdot)$ is an identity function that equals 1 if the condition is true and 0 otherwise. Based on Def. 2, a UQ method with coverage guarantee should satisfy $\text{Coverage}(\mathcal{D}_{\text{test}}) \geq 1 - \alpha$.*

One of the most straightforward frameworks to quantify uncertainty in $\hat{r}$ is conformal prediction (CP) (Shafer and Vovk, 2008), a frequentist distribution-free method relying on minimal assumptions. In the basic form for regression, i.e. estimation of $r_{\text{gt}}$, CP enables the following confidence intervals with theoretical guarantees.

**Proposition 4 (Conformalized Quantile Regression (CQR))** *(Romano et al., 2019) Given ground-truth $r_{\text{gt}}$, prediction $\hat{r}$ and the absolute error residual $e_r := |r_{\text{gt}} - \hat{r}|$, let $q_{e_r,\delta}$ denote the $\frac{n+1}{n}(1 - \delta)$ quantile of the instance-wise $e_r$ on a validation set $\mathcal{D}_{val}$ of size n. Then, with probability at least $1 - \delta$*

$$r_{\text{gt}} \in [\hat{r} - q_{e_r,\delta}, \hat{r} + q_{e_r,\delta}], \tag{4}$$

The biggest challenge of naïve CQR is that it does not allow for adaptivity. For example, in the case of TKR or NTR, it is important to take the tumor size into account, as it is much harder to annotate small objects. We therefore consider the adaptive CQR.

**Proposition 5 (Adaptive Conformalized Quantile Regression (ACQR))** *(Angelopoulos and Bates, 2021) Let $u_r > 0$ be an uncertainty measure of r, the instance-wise conformity score of the residual term is defined as $s_r := \frac{e_r}{u_r} = \frac{|r_{\text{gt}} - \hat{r}|}{u_r}$. Similar to Prop. 4, let $q_{s_r,\delta}$ denote the $\frac{n+1}{n}(1 - \delta)$ quantile of $s_r$ from $D_{\text{val}}$. Then, with probability at least $1 - \delta$*

$$r_{\text{gt}} \in [\hat{r} - u_r\, q_{s_r,\delta},\ \ \hat{r} + u_r\, q_{s_r,\delta}]. \tag{5}$$

*When $u_r = 1$, the score $s_r$ degrades to a residual term $e_r$, i.e. ACQR degrades to CQR.*

Despite the mathematical guarantees of ACQR, the choice of $u(x)$ remains non-trivial and requires domain expertise. This naturally limits its generalizability. In this paper, we follow the intuition that small tumors contain greater uncertainty and define $u(x)$ for RBBs in tumor cases as follows.

**Remark 6 (Uncertainty Measure in Tumors)** *In tumor-related applications, uncertainty is often characterized by tumor size. Consider $V_{\text{T}}$ being the tumor volume for sample x, and $V_{\text{T,max}}$ be the maximum tumor size that can be measured in a particular application. We then define $u(x)$ as*

$$u(x) = \lambda \left(1 - \frac{V_{\text{T}}}{V_{\text{T,max}} + \epsilon}\right),\ \text{with } \lambda = \frac{1}{2q_{s_r,\delta}}, \tag{6}$$

*for ACQR implementation, see derivation in Sec.B.1 in appendix.*

By Def. 1, the predicted ratio $\hat{r}$ is determined by the probability volumes predicted by the network. Since the network is not perfectly calibrated, quantifying the uncertainty in its predictions is closely tied to assessing volume bias.

**Definition 7 (Volume Bias (V-Bias))** *(Popordanoska et al., 2021) Given a segmentation model $g: \mathcal{Z} \to [0,1]$ that predicts the probability of $y \in \{0,1\}$, the volume bias is defined as:*

$$\text{V-Bias}(g) \coloneqq \mathbb{E}_{(z,y)\sim P}[g(z) - y]. \tag{7}$$

One can observe a direct connection between the V-bias and the residual in the definition of CP. It has been shown by (Popordanoska et al., 2021) that V-Bias is upper bounded by calibration error, which is mathematically defined as follows.

**Definition 8 (Calibration Error (CE))** *(Kumar et al., 2019) Given a model $g: \mathcal{Z} \to [0,1]$ that predicts the probability of $y \in \{0,1\}$, the calibration error is defined as:*

$$\text{CE}(g) \coloneqq \mathbb{E}_{(z,y)\sim P}[|g(z) - \mathbb{E}[y = 1 \mid g(z)]|], \tag{8}$$

The mentioned relationship between CE and V-bias was defined by Popordanoska et al. (2021) as follows.

**Proposition 9 (The Relationship of V-Bias and CE)** *(Popordanoska et al., 2021) Given segmentation model $g: \mathcal{Z} \to [0,1]$, the absolute value of volume bias is upper bound by the calibration error, i.e., $|\text{V-Bias}(g)| \leq \text{CE}(g)$.*

In the next section, we first derive a local RBB interval considering the bias and variance. Then, we extend the concept of Conformalized Quantile Regression to V-Bias to derive a miscalibration RBB bound. To provide a comprehensive analysis, we additionally discuss CE as an alternative upper bound.

## 3. CARE: Confidence-aware Ratio Estimation

**Overview.** In this section, we illustrate our insight of uncertainty analysis based on two key observations. The first observation is that the ratio estimator $\hat{r} = \frac{\bar{y}}{\bar{x}}$ is subject to instance-wise randomness, which we capture using statistical tools such as Markov's inequality to derive an *estimation-based interval*. The second observation is that the network is not perfectly calibrated, introducing a global, model-level error affecting both the numerator and denominator; this gives rise to the *calibration-based interval*. Combining these two sources yields the overall uncertainty bound.

**Estimation-based interval.** Van Kempen and Van Vliet (2000) provides an approximated theoretical result for ratio statistics. However, their derivation critically relies on the assumption that the addends in $\bar{x}$ and $\bar{y}$ are independent. Therefore, the result in Van Kempen and Van Vliet (2000) is not directly applicable in imaging analysis for violating spatial patterns. As a remedy, we construct Markov bounds as an estimation-based confidence interval for $\hat{r}$ using Markov inequality (Resnick, 2003). Although this approach leads to more conservative bounds, it avoids strong assumptions such as pixel independence, making it more applicable to image data.

**Proposition 10 (Estimation-based Confidence Interval)** *Given an estimator $\hat{r} = \frac{\bar{y}}{\bar{x}}$ of the fraction $r = \frac{\mathbb{E}[y]}{\mathbb{E}[x]}$ with random variables $x$ and $y$, it holds with at least $1 - \alpha$ probability that*

$$r \in [\hat{r} - \beta_{r,\alpha}, \ \hat{r} + \beta_{r,\alpha}], \tag{9}$$

*where $\beta_{r,\alpha} := \frac{\sqrt{\mathrm{SE}_{\hat{r}}}}{\sqrt{\alpha}}$ is the half-width of the bound, and $\mathrm{SE}_{\hat{r}} := \mathbb{E}\left[(\hat{r} - r)^2\right]$ is the expected squared error.*

Then we conduct a Taylor expansion of $\mathrm{SE}_{\hat{r}}$ to receive an approximation we can estimate in practice.

**Proposition 11** *Assume all central moments of the independently and identically distributed random variables $(x_1, y_1), \ldots, (x_n, y_n) \sim \mathbb{P}_{xy}$ in the estimator $\hat{r} = \frac{\bar{y}}{\bar{x}}$ exist, then we have*

$$\mathrm{SE}_{\hat{r}} = \frac{1}{n}\left(\frac{\mathrm{Var}(y)}{\mu_x} + \mathrm{Var}(x)\frac{\mu_y^2}{\mu_x^4} - 2\mathrm{Cov}(x,y)\frac{\mu_y}{\mu_x^3}\right) + O\left(\frac{1}{n^2}\right). \tag{10}$$

*Further, the estimator is:*

$$\widehat{\mathrm{SE}_{\hat{r}}} := \frac{1}{n}\left(\frac{\hat{\sigma}_y^2}{\bar{x}} + \frac{\hat{\sigma}_x^2 \bar{y}^2}{\bar{x}^4} - 2\frac{\hat{\sigma}_{xy}\bar{y}}{\bar{x}^3}\right), \tag{11}$$

*with the sample variances $\hat{\sigma}_x^2 = \frac{1}{n-1}\sum_i (x_i - \bar{x})^2$, $\hat{\sigma}_y^2 = \frac{1}{n-1}\sum_i (y_i - \bar{y})^2$, and sample covariance $\hat{\sigma}_{xy} = \frac{1}{n-1}\sum_i (x_i - \bar{x})(y_i - \bar{y})$. Under i.i.d. assumption, the estimator $\widehat{\mathrm{SE}_{\hat{r}}}$ is consistent, i.e., $\widehat{\mathrm{SE}_{\hat{r}}} \to \mathrm{SE}_{\hat{r}}$ in probability for $n \to \infty$. The proof is presented in the appendix B.2.*

**Calibration-based interval.** Then we analyze the second source of uncertainty: volume bias caused by miscalibration. Inspired by Prop. 4, we propose a fine-grained calibration-based confidence interval by considering the uncertainty of target (A) and RoI (B) regions separately, yielding asymmetric half-widths $\epsilon_{l,\delta}, \epsilon_{u,\delta}$ for lower and upper bounds. Unlike vanilla Conformalized Quantile Regression, where the analysis starts from the final $\hat{r}$, we adopt quantiles of $V_A$ and $V_B$ to give the calibration-based confidence interval of RBB, see Prop. 13 (appendix B.3). Combined with Prop. 10, we propose CARE (V-Bias), which requires minimum assumptions and gets rid of the dedicated uncertainty scores, compared with ACQR. As described in Prop. 9, V-Bias is upper bounded by the corresponding calibration error, *i.e.*, $|\text{V-Bias}(g_A)| \leq \mathrm{CE}(g_A)$, $|\text{V-Bias}(g_B)| \leq \mathrm{CE}(g_B)$. This motivates a more conservative interval named as CARE (ECE).

**Proposition 12 (Overall Confidence Interval)** *Assume we have a ratio estimator $\hat{r} = \frac{\sum_i g_A(z_{i,I})}{\sum_i g_B(z_{i,I})}$ for pixel measurements $\{z_{i,I}\}_{i=1}^n$ of an instance $I$ based on neural network outputs $g(z_{i,I}) = (g_A(z_{i,I}), g_B(z_{i,I}))$. Let $y_A$ and $y_B$ be the instance-wise target random variables used to form the target ratio $r = \frac{\mathbb{E}[y_A|I]}{\mathbb{E}[y_B|I]}$. Then, it holds with at least $1 - \alpha - \delta$ probability that*

$$r \in \left[\frac{\sum_i g_A(z_{i,I})}{\sum_i g_B(z_{i,I})} - \epsilon_{l,\delta} - \beta_{r,\alpha}, \ \frac{\sum_i g_A(z_{i,I})}{\sum_i g_B(z_{i,I})} + \epsilon_{u,\delta} + \beta_{r,\alpha}\right], \tag{12}$$

*where $\beta_{r,\alpha}$ is defined as in Prop. 10 and $\epsilon_{l,\delta}, \epsilon_{u,\delta}$ as in Prop. 13 (appendix B.3).*

The interval width $w = B_u - B_l$ measures the uncertainty level, as a result, a wide interval over thresholds alarms for manual examination. We perform a grid search of $\alpha$ and $\delta$, keeping $\alpha + \delta$ constant. The configuration yields the narrowest intervals that satisfy target coverage rates. In experiments (Sec. 4), we show empirically that CARE (V-Bais) achieves robust coverage and spans dynamically for different uncertainty levels. In addition, CARE (ECE) exhibits tighter bounds with comparable coverage *w.r.t* ACQR, without extra uncertainty assumption.

## 4. Experiments

### 4.1. Setup

**Datasets and models.** We evaluate our method on two brain tumor segmentation datasets: MSD-Task01 (Antonelli et al., 2022) and BraTS21 (Baid et al., 2021), both of which provide four segmentation labels (edema, necrosis, enhancing tumor, and background). The necrosis-to-tumor ratio (NTR) is defined as $\frac{V_N}{V_T}$, *i.e.* the ratio between the necrotic volume $V_N$ and the whole tumor volume $V_T$ (edema, necrosis, and enhancing regions). We additionally include KiTS23 (Myronenko et al., 2023), a CT dataset of 489 kidney volumes, where the tumor-to-kidney ratio (TKR) is defined as $\frac{V_T}{V_{Kidney}}$. To predict these biomarkers from segmentation outputs, we train nnUNet (Isensee et al., 2021), nnFormer (Zhou et al., 2021), and UNETR++ (Zhou et al., 2021) using a nested five-fold cross-validation. The predicted ratio $\hat{r}$ and labeled ratio $r_{gt}$ are computed from Def. 1.

**Uncertainty Quantification baselines.** To control the confidence level to be $C = 0.68$, we adopt a quantile for CARE, CQR, ACQR and sampling-based methods. We also implement two Bayesian methods: ensemble and dropout. Due to the expensive inference, we report their $3\sigma$ confidence intervals. More implementation details in appendix (Sec.A.1).

**Metrics.** Recap Def. 2, a sound confidence interval can be measured by: (1) *Coverage rate*: The proportion of samples whose ground-truth values fall within the estimated confidence intervals (Def. 3). A sound confidence interval should achieve the prescribed confidence level in terms of coverage rate, while remaining as tight as possible. (2) *Adaptiveness*: The interval width should reflect the difficulty of prediction. For ratio-based biomarkers, the interval width is expected to increase as the tumor size decreases, since small tumors are generally harder to segment and tend to exhibit larger prediction errors.

Notably, there exists an intrinsic trade-off between *coverage rate* and *tightness*: a naively loose and non-adaptive interval can trivially satisfy the coverage requirement, but becomes uninformative for practical use. Therefore, we evaluate UQ methods using both metrics jointly: We first identify methods that satisfy the desired coverage level (Tab. 1), and then compare their adaptiveness and tightness among the qualified methods (Fig. 3).

### 4.2. Results

**Coverage guarantee.** As described in Sec.1, a conservative confidence interval achieves coverage probability higher than the nominal confidence level, *i.e.*, achieving over 68% coverage when aiming for 68% confidence level. We report coverage rate (%) of different UQ methods at 0.68 confidence level in Table 1, which measures *the proportion of samples*

Table 1: Comparison of the coverage guarantee on MSD-Task01 dataset ($C = 0.68$). We report the overall coverage rate (%) on test-set ($\pm$: error bar). CARE always satisfies the desired confidence level without being overconservative.

| Coverage (%) | nnUNet$_{2d}$ | nnUNet$_{3d}$ | nnFormer | UNETR++ |
|---|---|---|---|---|
| **Ensemble ($1\sigma$)** | $6.21_{\pm0.36}$ | $7.54_{\pm0.78}$ | $6.72_{\pm0.56}$ | $8.24_{\pm0.74}$ |
| **Dropout ($1\sigma$)** | $5.78_{\pm0.43}$ | $7.12_{\pm0.66}$ | $6.23_{\pm0.71}$ | $8.01_{\pm0.86}$ |
| **CQR** | $72.11_{\pm1.90}$ | $67.23_{\pm3.88}$ | $67.92_{\pm1.59}$ | $65.76_{\pm2.11}$ |
| **ACQR** | $94.22_{\pm1.89}$ | $94.22_{\pm2.88}$ | $91.78_{\pm1.39}$ | $93.15_{\pm1.91}$ |
| **CARE (ECE)** | $94.22_{\pm0.99}$ | $93.61_{\pm0.71}$ | $87.94_{\pm0.97}$ | $89.58_{\pm1.02}$ |
| **CARE (V-Bias)** | $93.61_{\pm1.14}$ | $86.60_{\pm1.49}$ | $81.92_{\pm1.31}$ | $76.43_{\pm2.21}$ |

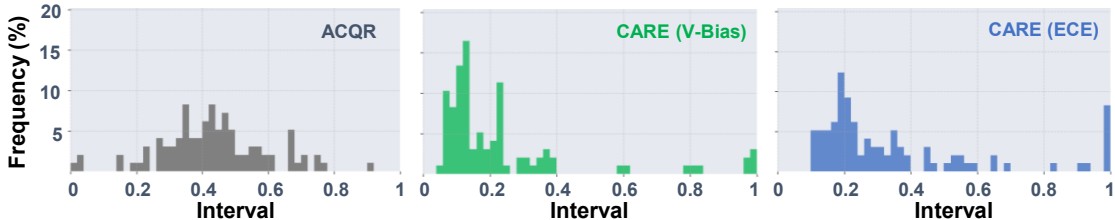

(a) **Interval distributions on MSD-Task01 dataset.**

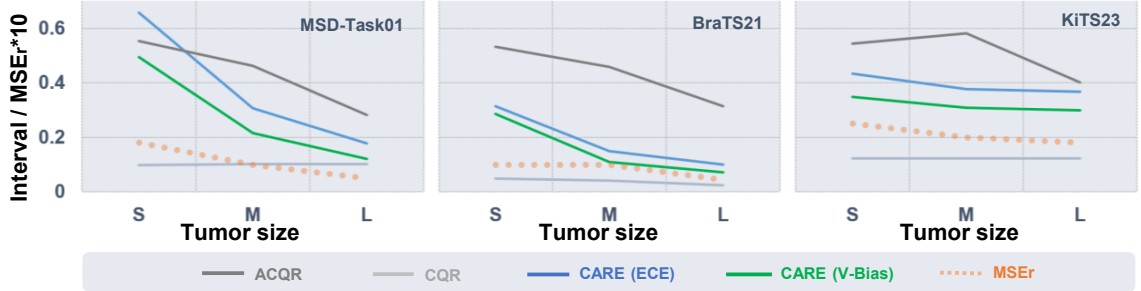

(b) **Interval widths stratified by tumor-size on 3 datasets.**

Figure 3: Comparison of adaptiveness on nnUNet$_{3d}$ ($C = 0.68$). (a) The frequency histogram of NTR intervals in test-set. ACQR's intervals lie frequently around the middle area, while CARE has tighter bounds generally. (b) The average interval width in three groups categorized by tumor sizes. Intuitively, interval width should reflect MSE$_r$ tendency. Compared with the indistinguishable CQR and overconservative ACQR, CARE varies appropriately wider for small tumors (hard samples) and tighter for large ones (simple).

*whose true values fall within the confidence intervals.* Empirically, our intervals show higher likelihoods of satisfying the prescribed confidence level of 0.68 compared with sampling-based methods and CQR. Notably, the Bayesian methods (dropout and ensemble) show poor coverage due to a lack of an appropriate prior. Considering the suboptimal performance of

sampling-based methods, our following comparison focuses on two methods with the coverage guarantee: ACQR and CARE.

**Adaptiveness.** Beyond achieving the guaranteed coverage rate, the confidence interval should be sample-adaptive to identify unreliable predictions effectively. We demonstrate this capability by examining the "dataset-level interval" distribution of MSD-Task01 in Fig. 3 (a). As observed, most ACQR intervals lie around 0.4, showing overall conservative bounds. In contrast, our method produces intervals that adapt with the tumor size (see per-sample visualizations of intervals in appendix, Fig. B). Furthermore, the uncertainty should correlate appropriately with segmentation difficulty. For instance, small tumors are hard to detect and segment for their small size, low contrast and susceptibility to noise. Empirically, hard samples with small sizes or blurry boundaries tend to yield erroneous predictions (large mean squared error), necessitating wider intervals to ensure coverage. To validate this adaptive behavior, we present fine-grained analysis of $MSE_r$ (error measures) and interval width (uncertainty measures) in Fig. 3 (b), including NTR in MSD-Task01, NTR in BraTS21 and TKR in KiTS23. We stratify tumors into small (S), medium (M), and large (L) categories based on the $\frac{1}{3}$ and $\frac{2}{3}$ quantiles of tumor sizes in test-set. As illustrated, our interval widths are associated with the segmentation difficulty: smaller, more challenging tumors receive wider intervals, while larger, easier-to-segment tumors receive narrower intervals. In comparison, CQR is unable to distinguish different uncertainty levels, which prevents it from identifying high-risk predictions. Although both ACQR and CARE shrink their intervals for larger tumors, the extremely wide ACQR interval for small tumors reduces sensitivity to tumor-specific variations.

### 4.3. Further Study

**Tunability and robustness.** To demonstrate tunability and robustness across different confidence levels, we report NTR coverage rates on varying confidence thresholds in Fig. 4 (a). The coverage rate is expected to increase proportionally with the increased confidence level. However, CQR struggles to achieve the desired confidence and ACQR tends to be overconservative as the upper bound CARE (ECE).

**Temperature effects.** Then, we report interval width under different temperature parameters in Fig. 4 (b), to observe the effect of post-hoc calibration on confidence measures and CARE. The ECE of necrosis and tumor ($ECE_{N, T}$) reflects the miscalibration degree and the average interval width of CARE (ECE) works as the uncertainty measure. For illustration, we scale up ECE by 100. As observed, both ECE and our interval width decrease as the temperature increases. This indicates that CARE becomes tighter for a well-calibrated model, and vice versa.

**Uncertainty decomposition.** As described in Sec. 3, we decompose uncertainty into miscalibration and intrinsic bias of ratio estimation. We analyze their contribution empirically by ablation on interval widths. Specifically, we calculate estimation-based ($I_{Est}$), V-Bias-based ($I_{V-Bias}$) and ECE-based ($I_{ECE}$) confidence intervals respectively. The results in Fig. 4 (c) show that the miscalibration-based intervals, $I_{V-Bias}$ and $I_{ECE}$, are much wider than $I_{Est}$, indicating that model miscalibration is the primary uncertainty source in ratio estimation.

**Uncertainty measure** $u(x)$ **in ACQR.** The coverage and adaptiveness of ACQR rely heavily on a dedicated $u(x)$, as discussed in appendix A.3. In comparison, CARE provides a more straightforward and robust solution, through a clean and principled construction.

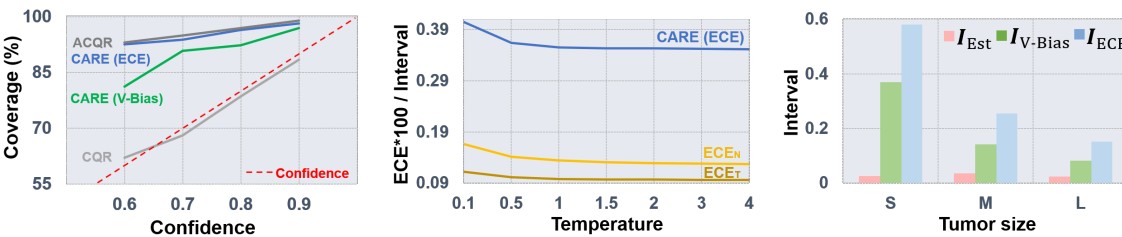

(a) **Confidence levels.**    (b) **Temperature effects.**   (c) **Uncertainty ablation.**

Figure 4: Further study on MSD-Task01 and nnUNet$_{3d}$ ($C = 0.68$). (a) CARE satisfies the desired confidence levels consistently. (b) When the temperature moves towards better calibration (ECE ↓), our interval becomes narrower (Interval ↓). (c) Miscalibration is the main contributor to the overall uncertainty, since the ECE-only interval $I_{\text{ECE}}$ takes the dominant portion of the overall interval $I_{\text{O}}$.

**The size of** $D_{\text{val}}$**.** In our experiments, the validation set of MSD-Task01 has 38 samples, which is already small. Yet, we randomly sample a smaller set from $D_{\text{val}}$ in Tab. 2. CARE maintains the confidence level and adaptive intervals even with 10 samples. Moreover, the same distribution assumption is hardly guaranteed for small $D_{\text{val}}$. Therefore, Tab. 2 indicates our robustness towards smaller validation sets and domain shift.

**Grid search.** As shown in Tab. 3, our grid search over the combination of $(\alpha, \delta)$ yields the minimal interval width under the coverage guarantee ($C = 0.68$). Theoretically, the coverage rate is guaranteed to over the desired $C$ regardless of the choice of $(\alpha, \delta)$ combination under $\alpha + \delta = 1 - C$. Nevertheless, to be practical for an informative alarm, we conduct a grid search to find the narrowest confidence intervals.

Table 2: Sizes of the validation set on MSD-Task01 and nnUNet$_{3d}$ ($C = 0.68$).

| #Samples | Coverage (%) | Interval (S/M/L) |
|---|---|---|
| 10 | $85.78_{\pm 1.68}$ | $0.51_{\pm 0.09}/0.26_{\pm 0.10}/0.14_{\pm 0.07}$ |
| 20 | $86.10_{\pm 1.39}$ | $0.56_{\pm 0.10}/0.28_{\pm 0.12}/0.13_{\pm 0.08}$ |
| 30 | $85.92_{\pm 1.53}$ | $0.42_{\pm 0.09}/0.20_{\pm 0.09}/0.14_{\pm 0.08}$ |

Table 3: Grid search on MSD-Task01 and nnUNet$_{3d}$ ($C = 0.68$).

| $(\alpha, \delta)$ | Coverage (%) | Interval (S/M/L) |
|---|---|---|
| (0.02, 0.30) | $91.75_{\pm 1.96}$ | $0.50_{\pm 0.11}/0.26_{\pm 0.09}/0.16_{\pm 0.07}$ |
| (0.30, 0.02) | $100.00_{\pm 0.00}$ | $0.98_{\pm 0.07}/0.76_{\pm 0.08}/0.41_{\pm 0.10}$ |
| Grid | $86.60_{\pm 1.49}$ | $0.49_{\pm 0.09}/0.21_{\pm 0.07}/0.12_{\pm 0.05}$ |

**Segmentation metrics.** The Dice score and IoU of the tumor are reported in Tab. 4. As tumors grow, the ratio estimation and segmentation performance are declining at different rates, since the ratio estimator accounts for two variables (tumor and necrosis). The overall degradation tendency of segmentation (Dice and IoU in Tab. 4) and ratio estimation (MSE in Fig. 3 (b)) indicates a relationship between tumor size and prediction difficulty. Therefore, our adaptiveness on interval widths is justified.

**Inference time.** In consistent with the main paper, we measure the inference time on a single A100 GPU. Tab. 5 shows the extra inference time of CARE, which is negligible regarding the segmentation duration.

Table 4: Segmentation results.

| Tumor$_{\text{size}}$ | Dice | IoU |
|---|---|---|
| S | 0.65 | 0.53 |
| M | 0.76 | 0.65 |
| L | 0.77 | 0.65 |

Table 5: Inference time.

| Dataset | Seg. (ms) | CARE (ms) |
|---|---|---|
| MSD-Task01 | $1680.69_{\pm 1.03}$ | $+5.21_{\pm 0.12}$ |
| BraTS21 | $1681.69_{\pm 1.10}$ | $+5.25_{\pm 0.11}$ |

## 5. Related Work

**Uncertainty quantification** provides statistical methods to estimate prediction uncertainty. *Adaptive Conformalized Quantile Regression (ACQR)* (Vovk et al., 1999, 2005) constructs prediction intervals that guarantee valid coverage under finite samples, without any distributional assumptions. Its key strength is the distribution-free nature and finite-sample validity, providing strong theoretical guarantees regardless of the base predictive model. *Resampling methods* are non-parametric techniques for estimating the sampling distribution of a statistic, applicable when the underlying distribution is unknown or difficult to derive. Specifically, *Bootstrapping* (Mooney et al., 1993; Freedman, 1981) repeatedly samples $N$ data points with replacement from the original data, whereas *subsampling* (Politis and Romano, 1994) takes a subset of the original data without replacement, repeating the process multiple times to construct an empirical distribution of the statistic. *Bayesian methods* achieve robust segmentation by averaging multiple predictions, using techniques like deep ensemble (Lakshminarayanan et al., 2017) and Monte Carlo dropout (Srivastava et al., 2014).

**Calibration error** estimation has attracted extensive research attention (Kull and Flach, 2015; Vaicenavicius et al., 2019; Kumar et al., 2019; Zhang et al., 2020; Popordanoska et al., 2022; Gruber and Buettner, 2022). In medical segmentation, classwise and canonical calibration error are used to evaluate per-structure and overall calibration levels. Derived from individual channel masks, the classwise CE in multi-class segmentation simplifies to binary CE for each channel. In addition, (Popordanoska et al., 2021) proves that the absolute value of volume bias (V-Bias) is upper-bounded by CE. Many calibration methods like temperature scaling (Guo et al., 2017) and isotonic regression (Zadrozny and Elkan, 2002) have been proposed to improve the calibration of classification scores. However, no previous work analyzes how miscalibration affects downstream ratio-based estimates.

## 6. Conclusion

We propose CARE, a confidence-aware framework for estimating ratio-based biomarkers from segmentation network outputs. Our method addresses a common limitation of prior works that focus solely on point estimates without confidence guarantees. We disentangle two key sources of uncertainty, *i.e.* network prediction error and statistical bias. Our empirical findings highlight that miscalibration is a dominant contributor to uncertainty. Our framework offers several practical advantages: it operates as a model-agnostic plugin module, provides sample-level adaptive uncertainty estimates in a single forward pass without requiring multiple sampling, and allows users to flexibly adjust confidence levels. In summary, this work represents an important step toward trustworthy deployment of deep learning in clinical settings by providing practitioners with both accurate biomarker estimates and reliable confidence bounds.

Despite the practical advantages, our work has several limitations. First, we assume that the validation and test sets are drawn from the same distribution. Although it is standard in supervised learning settings, but may not hold under domain shifts. In practice, domain shifts arise due to differences in scanners, acquisition protocols, or patient populations. As a result, our confidence interval may not remain valid in these scenarios. Addressing this challenge with label-free calibration error estimators (e.g. Wang et al. (2020); Popordanoska et al. (2024)) is a promising direction for future work. Second, the quality of the calibration of the underlying segmentation network has an impact on the tightness of the derived confidence intervals. Specifically, when the calibration error is large, the resulting confidence intervals may become overly conservative. Improving calibration in segmentation networks would directly translate into narrower, more informative confidence intervals within our approach. Finally, while our framework shows good performance on public datasets, clinical validation is needed to assess its real-world impact on decision-making and patient outcomes.

Despite limitations, we have shown the first confidence-aware method for estimating confidence intervals in imaging-based ratio biomarkers. Compared with existing baselines, our method yields intervals that are both tight and adaptive. We believe this provides a solid foundation for the next generation of AI systems capable of propagating uncertainty throughout the entire deep-learning-based biomarker estimation pipeline.

### Acknowledgments

This research received funding from the Flemish Government (AI Research Program) and the Research Foundation Flanders (FWO) through project number G0G2921N. Aleksei was supported by Sigrid Juselius Foundation and the Finnish Research Council (Profi6 336449 funding program), the strategic funding of the University of Oulu. We acknowledge EuroHPC JU for awarding the project ID EHPC-AIF-2025SC02-042 access to the EuroHPC supercomputer LEONARDO, hosted by CINECA (Italy) and the LEONARDO consortium.

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

# Appendix

A recap of our idea is shown in Fig. A. In Appendix A, we further illustrate experimental details and present additional experimental results, relevant to our methodology and in support of the main paper. In Appendix B, we offer the proofs of propositions in the main paper. Finally, we give related work in Appendix C.

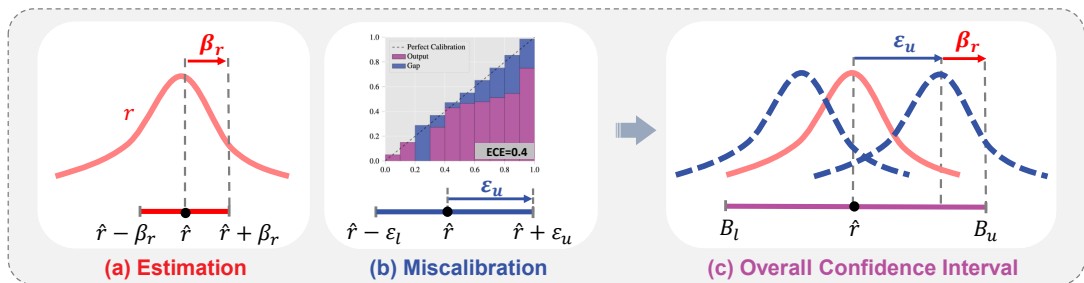

Figure A: Our confidence interval considering estimation and miscalibration. **(a)** shows Markov bounds from the estimator. **(b)** illustrates the prediction offset $\epsilon_{l,u}$ due to miscalibration. **(c)** is the overall confidence interval $r \in [B_l, B_u]$.

## Appendix A. Experiments

### A.1. Experimental Details

**Datasets.** MSD-Task01 (Antonelli et al., 2022) and BraTS21 (Baid et al., 2021) include 484 and 1251 MRI volumes respectively, with four modalities (T1, T2, T1ce, FLAIR) and four annotations (edema, necrosis, enhancing tumor, background). KiTS23 (Myronenko et al., 2023) is a CT dataset of 489 kidney volumes, with four annotations (tumor, kidney, cyst, background). A nested five-fold cross-validation is used for all datasets. In the outer loop, four folds are used for training and validation, and the remaining one fold for testing. Within the inner loop, 10% of the training data is held out as a validation set $\mathcal{D}_{\mathrm{val}}$ to estimate the quantile of V-Bias and ECE.

**Segmentation models.** We conduct experiments using nnUNet (Isensee et al., 2021), nnFormer (Zhou et al., 2021) and UNETR++ (Zhou et al., 2021). All models are trained using cross-entropy (XE) (Bishop and Nasrabadi, 2006) and soft Dice (SD) (Milletari et al., 2016) loss, label-based supervision and softmax activation under a single A100 GPU.

**Implementation details.** For conformal prediction (Vovk et al., 1999; Papadopoulos et al., 2002), we take the $((\frac{n+1}{n}) \cdot 0.68)$ quantile of absolute error residual $e_r$ from the validation set as the half-width (Prop. 4), while for CARE, we adopt dynamic V-Bias quantiles or ECE quantiles by conducting a grid search under the constraint of $1 - \alpha - \beta = 0.68$ (Prop. 12). For Bayesian methods, conducting numerous forward passes to estimate a "tunable" quantile is computationally impractical; thus, we report the results of three standard deviations ($3\sigma$). Ensemble intervals are obtained from $K$ models trained with different seeds, and dropout intervals come from $K$ forward passes ($K = 20$). To implement sampling-based methods, we repeatedly sample pixels from an instance and calculate its ratio estimate for 100 times, then adopt the $[0.16, 0.84]$ quantile from 100 repetitions as the 0.68 confidence level. Specifically, for a volume of $N$ pixels, we take $0.1N$ random pixels each time without replacement for subsampling (Politis and Romano, 1994), and sample $N$ pixels with replacement each time for bootstrapping (Mooney et al., 1993).

**A.2. Coverage Guarantee and Adaptiveness**

In the main paper, we just give the overall confidence intervals histogram in Fig. 3 (a). To provide a more comprehensive, "bird-eye" view of our method's behavior, we extend this analysis to the whole test samples in Fig. B, where we plot $\hat{r}$ and the confidence intervals under four methods. For clarity, the sample indices are omitted. As shown in Fig. B (a), CQR has symmetric half bandwidths and nearly uniform interval widths, which disables the identification function. ACQR (b) provides adaptive intervals while behaving overconservative. In comparison, our CARE shows adaptive intervals with desired distinction, which is particularly important in clinical settings to provide a reliable and informative reference.

**A.3. Ablation on $u(x)$**

In Remark 6, we assume a known maximum tumor size and set a scaling factor $\lambda = \frac{1}{2q_{s_r,\delta}}$ for the uncertainty measure $u(x)$, which extends the ACQR distribution across $[0, 1]$. Here, we show two variants of $u(x)$: (i) without $\lambda$, a less well-designed $u(x)$; (ii) with voxel size $V$, *i.e.* $u(x) = 1 - \frac{V_T}{V}$, assuming unknown max. tumor size. Since the tumor size is much smaller than the whole voxel size, we adopt $\frac{1}{8}V$ as the denominator for the second variant. Following the format in Fig. 3 (b) and Fig. B, we report these results in Fig. C. Compared with our implementation in the main paper, both two variants are less adaptive while yielding narrow intervals. The prior of the voxel size is easier to obtain than maximum tumor size. However,

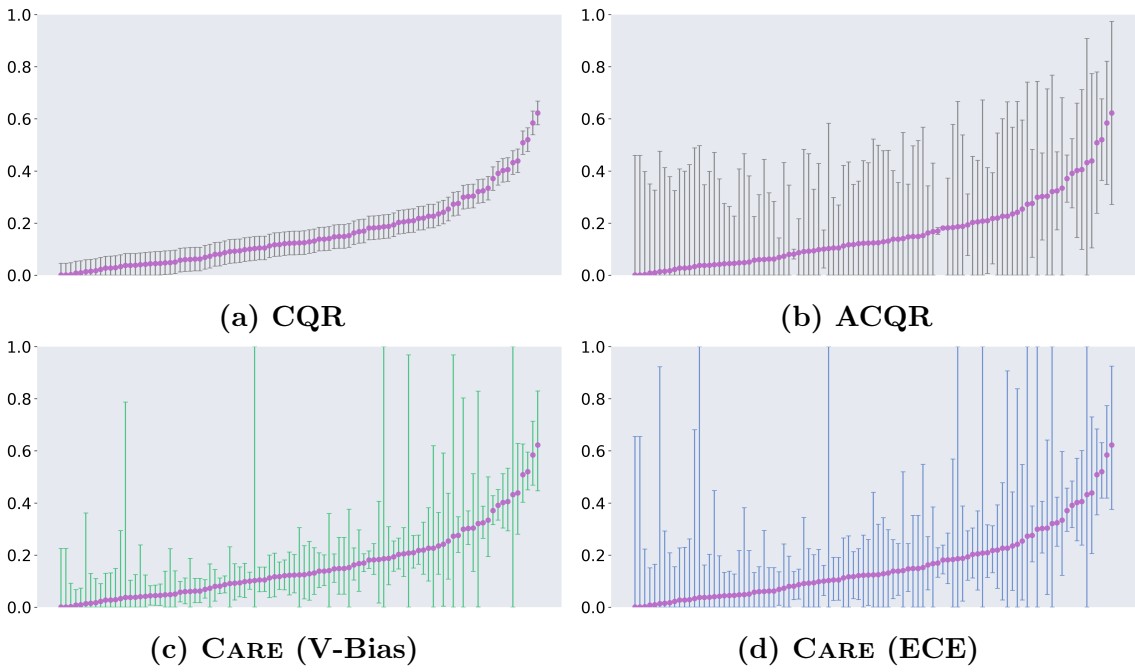

Figure B: Visualization of our confidence intervals on MSD and nnUNet$_{\mathrm{3d}}$. The x-axis represents all test samples sorted by predicted ratio $\hat{r}$, and the y-axis displays the valid range of ratio estimates.

the common but less informative prior "dilutes" the adaptiveness of ACQR, for its nearly uniform intervals. Fig. C further indicates the significant role of $u(x)$ on ACQR performance, which is also the drawback for wider application.

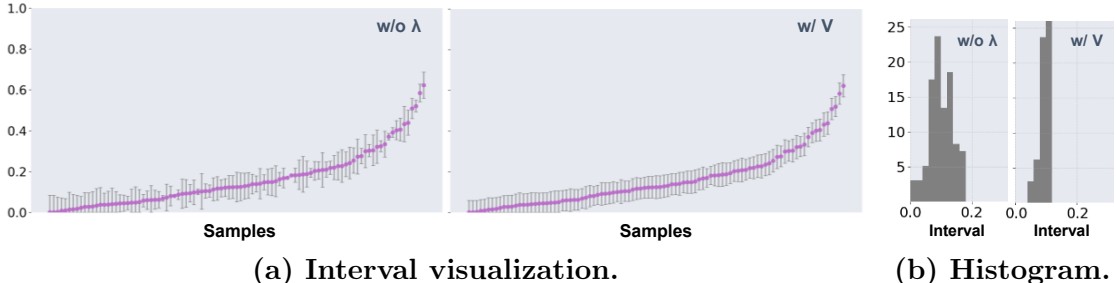

(a) Interval visualization.       (b) Histogram.

Figure C: Ablation study on $u(x)$. "w/o $\lambda$" means $u(x) = 1 - \frac{V_T}{V_{T,.max}}$; "w/ V" means $u(x) = 1 - \frac{V_T}{V}$. Both methods provide limited spans of confidence intervals, where all interval widths are below 0.2.

## Appendix B. Proofs

In this section, we first give the corresponding proof of the scaling factor $\lambda$(B.1) mentioned in Remark. 6. Then we show the proof of Markov bounds (B.2) and miscalibration bounds (B.3) mentioned in Sec. 3. Finally, we derive a debiased estimator in Sec. B.4.

### B.1. Uncertainty Measures in Tumors

Recall that in ACQR, the confidence interval for a ratio-based biomarker $r(x)$ is defined as $\hat{r}(x) \pm u(x)q_{s_r,\delta}$, where $q_{s_r,\delta}$ is the $(\frac{n+1}{n})\delta$-quantile of the score $s_r$. We choose the uncertainty measure $u(x) = \lambda \left(1 - \frac{V_T}{V_{T,max}+\epsilon}\right)$.

The maximum possible width $I_{max}$ occurs when $V_T \to 0$:

$$I_{max} = 2 \cdot u_{max} q_{s_r,\delta} = 2 \cdot \lambda q_{s_r,\delta}. \tag{13}$$

As the ratio is always in $[0, 1]$:

$$2 \cdot \lambda q_{s_r,\delta} = 1 \implies \lambda = \frac{1}{2q_{s_r,\delta}}. \tag{14}$$

### B.2. Markov Bounds

(Van Kempen and Van Vliet, 2000) provides a confidence interval of the ratio estimator $\frac{\bar{y}}{\bar{x}}$ based on asymptotic normal assumptions and by using the variance $\sigma_r^2 := \text{Var}\left(\frac{\bar{y}}{\bar{x}}\right)$. However, adopting their results assumes that all pixels are independently and identically distributed, i.e., $(x_1, y_1), \ldots, (x_n, y_n) \overset{\text{i.i.d.}}{\sim} \mathbb{P}_{xy}$. In addition, they perform multiple approximation steps, and some approximations happen within the square operator. How the estimator behaves facing a violation of these assumptions is unknown in practice. In the following, we prove the alternative approach, we proposed in the main paper, which is based on Markov's inequality

([Resnick, 2003](#)). For conciseness, the "$\approx$" sign is avoided while we directly note the remainder terms for a rigorous analysis.

To avoid relying on any distribution assumptions, we construct a confidence interval via Markov's inequality for the estimator $\hat{r} = \frac{\bar{y}}{\bar{x}}$ and target $r = \frac{\mu_y}{\mu_x}$. We have

$$\mathbb{P}\left(|\hat{r} - r| \geq k\sqrt{\mathrm{SE}_{\hat{r}}}\right) = \mathbb{P}\left((\hat{r} - r)^2 \geq k^2 \mathrm{SE}_{\hat{r}}\right) \leq \frac{1}{k^2} \tag{15}$$

with the squared error $\mathrm{SE}_{\hat{r}} := \mathbb{E}\left[(\hat{r} - r)^2\right]$. We emphasize that in general $\sqrt{\mathrm{SE}_{\hat{r}}} \neq \sigma_r$.

In main paper, we denote $\alpha := \frac{1}{k^2}$ as the non-coverage probability. For instance, adopting the $1 - \alpha = 75\%$ confidence interval corresponds to $\alpha = \frac{1}{k^2} = 0.25$ or $k = 2$. Then the half-width of confidence interval is $2\sqrt{\mathrm{SE}_{\hat{r}}}$, *i.e.*, two times the root squared error. This is more conservative than using the normal assumption, but requires no distribution assumption.

Now, we compute the squared error via Taylor expansion ([Spivak, 2006](#)). First, note that

$$\mathrm{SE}_{\hat{r}} = \mathbb{E}\left[\left(\frac{\bar{y}}{\bar{x}} - \frac{\mu_y}{\mu_x}\right)^2\right] = \mathbb{E}\left[\frac{\bar{y}^2}{\bar{x}^2}\right] - 2\frac{\mu_y}{\mu_x}\mathbb{E}\left[\frac{\bar{y}}{\bar{x}}\right] + \frac{\mu_y^2}{\mu_x^2}. \tag{16}$$

We perform a Taylor expansion of $\frac{\bar{y}^2}{\bar{x}^2}$ around $\frac{\mu_y}{\mu_x}$ to compute its expectation:

$$\begin{aligned}
\frac{\bar{y}^2}{\bar{x}^2} = {} & \frac{\mu_y^2}{\mu_x^2} + 2\left(\bar{y} - \mu_y\right)\frac{\mu_y}{\mu_x^2} - 2\left(\bar{x} - \mu_x\right)\frac{\mu_y^2}{\mu_x^3} \\
& + \left(\bar{y} - \mu_y\right)^2\frac{1}{\mu_y} + 3\left(\bar{x} - \mu_x\right)^2\frac{\mu_y^2}{\mu_x^4} - 4\left(\bar{y} - \mu_y\right)\left(\bar{x} - \mu_x\right)\frac{\mu_y}{\mu_x^3} \\
& + \sum_{i,j:\ i+j\geq 3}\left(\bar{y} - \mu_y\right)^i\left(\bar{x} - \mu_x\right)^j\frac{\partial^{i+j}}{\partial^i\mu_y\partial^j\mu_x}\frac{\mu_y^2}{\mu_x^2}
\end{aligned} \tag{17}$$

from which follows

$$\begin{aligned}
\mathbb{E}\left[\frac{\bar{y}^2}{\bar{x}^2}\right] = {} & \frac{\mu_y^2}{\mu_x^2} + \frac{\mathrm{Var}\left(\bar{y}\right)}{\mu_y} + 3\,\mathrm{Var}\left(\bar{x}\right)\frac{\mu_y^2}{\mu_x^4} - 4\,\mathrm{Cov}\left(\bar{x}, \bar{y}\right)\frac{\mu_y}{\mu_x^3} \\
& + \sum_{i,j:\ i+j\geq 3}\mathbb{E}\left[\left(\bar{x} - \mu_x\right)^i\left(\bar{y} - \mu_y\right)^j\right]\frac{\partial^{i+j}}{\left(\partial\mu_x\right)^i\left(\partial\mu_y\right)^j}\frac{\mu_y^2}{\mu_x^2}.
\end{aligned} \tag{18}$$

Assuming $(x_1, y_1), \ldots, (x_n, y_n) \sim \mathbb{P}_{xy}$ are i.i.d. further simplifies the terms, like in the following. Markov's inequality does not require this assumption, so a violation does not invalidate our approach. Then, it holds that

$$\mathrm{Var}\left(\bar{x}\right) = \frac{1}{n}\mathrm{Var}\left(x\right), \quad \mathrm{Var}\left(y\right) = \frac{1}{n}\mathrm{Var}\left(y\right), \quad \mathrm{Cov}\left(\bar{x}, \bar{y}\right) = \frac{1}{n}\mathrm{Cov}\left(x, y\right). \tag{19}$$

Further, for all $a = 1, \ldots n$ let $z_{k,a} = x_a$ and $\mu_{z_k} = \mu_x$ if $1 \leq k \leq i$, and $z_{k,a} = y_a$ and $\mu_{z_k} = \mu_y$ if $i < k \leq m := i + j$. Then

$$
\begin{aligned}
\mathbb{E}&\left[ (\bar{x} - \mu_x)^i (\bar{y} - \mu_y)^j \right] \\
&= \frac{1}{n^{i+j}} \mathbb{E}\left[ \left( \sum_{a=1}^{n} x_a - \mu_x \right)^i \left( \sum_{a=1}^{n} y_a - \mu_y \right)^j \right] \\
&= \frac{1}{n^m} \mathbb{E}\left[ \prod_{k=1}^{m} \left( \sum_{a=1}^{n} z_{k,a} - \mu_{z_k} \right) \right] \\
&= \frac{1}{n^m} \sum_{l=1}^{m} \sum_{a_l=1}^{n} \mathbb{E}\left[ \prod_{k=1}^{m} \left( z_{k,a_k} - \mu_{z_k} \right) \right]
\end{aligned}
\tag{20}
$$

For all $a_k$ holds that $\mathbb{E}\left[ \prod_{k=1}^{m} (z_{k,a_k} - \mu_{z_k}) \right] = 0$ if there exists any non-duplicate index value, due to independence. It follows that we can reduce the number of indices by at least half, which reduces the number of addends by a polynomial:

$$
\begin{aligned}
\frac{1}{n^m} &\underbrace{\sum_{l=1}^{m} \sum_{a_l=1}^{n} \mathbb{E}\left[ \prod_{k=1}^{m} \left( z_{k,a_k} - \mu_{z_k} \right) \right]}_{n^m \text{ addends}} \\
&= \frac{1}{n^m} \underbrace{\sum_{l=1}^{\lfloor m/2 \rfloor} \sum_{a_l=1}^{n} \mathbb{E}\left[ \prod_{k=1}^{m} \left( z_{k,a_k} - \mu_{z_k} \right) \right]}_{n^{\lfloor m/2 \rfloor} \text{ addends}} \\
&= \frac{1}{n^{\lceil m/2 \rceil}} \frac{1}{n^{\lfloor m/2 \rfloor}} \underbrace{\sum_{l=1}^{\lfloor m/2 \rfloor} \sum_{a_l=1}^{n} \mathbb{E}\left[ \prod_{k=1}^{m} \left( z_{k,a_k} - \mu_{z_k} \right) \right]}_{=:C_{ij}}.
\end{aligned}
\tag{21}
$$

Note that $C_{ij} \in [-B_m, B_m]$ with $B_m := \max_{\{i,j=0,\ldots m | i+j \leq m\}} \left| \mathbb{E}\left[ (x - \mu_x)^i (y - \mu_y)^j \right] \right|$, therefore, the convergence rate depends not only on the data size $n$ but also on how the moments grow with $m$.

Using Eqn. 19 and Eqn. 21 gives

$$
\begin{aligned}
\mathbb{E}\left[ \frac{\bar{y}^2}{\bar{x}^2} \right] &= \frac{\mu_y^2}{\mu_x^2} + \frac{\operatorname{Var}(y)}{n\mu_y} + 3\operatorname{Var}(x) \frac{\mu_y^2}{n\mu_x^4} - 4\operatorname{Cov}(x,y)\frac{\mu_y}{n\mu_x^3} \\
&\quad + \sum_{i,j:\ i+j \geq 3} \frac{1}{n^{\lceil (i+j)/2 \rceil}} C_{ij} \frac{\partial^{i+j}}{(\partial\mu_x)^i (\partial\mu_y)^j} \frac{\mu_y^2}{\mu_x^2}.
\end{aligned}
\tag{22}
$$

Similarly, we use Taylor expansion for $\frac{\bar{y}}{\bar{x}}$ around $\frac{\mu_y}{\mu_x}$ to get

$$
\begin{aligned}
\frac{\bar{y}}{\bar{x}} = &\frac{\mu_y}{\mu_x} + (\bar{y} - \mu_y)\frac{1}{\mu_x} - (\bar{x} - \mu_x)\frac{\mu_y}{\mu_x^2} \\
&+ 0 + (\bar{x} - \mu_x)^2\frac{\mu_y}{\mu_x^3} - (\bar{y} - \mu_y)(\bar{x} - \mu_x)\frac{1}{\mu_x^2} \\
&+ \sum_{i,j:\ i+j\geq 3}(\bar{y} - \mu_y)^i(\bar{x} - \mu_x)^j\frac{\partial^{i+j}}{(\partial\mu_x)^i(\partial\mu_y)^j}\frac{\mu_y}{\mu_x},
\end{aligned}
\tag{23}
$$

which results in

$$
\begin{aligned}
\frac{\mu_y}{\mu_x}\mathbb{E}\left[\frac{\bar{y}}{\bar{x}}\right] = &\frac{\mu_y^2}{\mu_x^2} + \mathrm{Var}(\bar{x})\frac{\mu_y^2}{\mu_x^4} - \mathrm{Cov}(\bar{y}, \bar{x})\frac{\mu_y}{\mu_x^3} \\
&+ \sum_{i,j:\ i+j\geq 3}\mathbb{E}\left[(\bar{y} - \mu_y)^i(\bar{x} - \mu_x)^j\right]\frac{\mu_y}{\mu_x}\frac{\partial^{i+j}}{(\partial\mu_x)^i(\partial\mu_y)^j}\frac{\mu_y}{\mu_x} \\
= &\frac{\mu_y^2}{\mu_x^2} + \mathrm{Var}(x)\frac{\mu_y^2}{n\mu_x^4} - \mathrm{Cov}(x, y)\frac{\mu_y}{n\mu_x^3} \\
&+ \sum_{i,j:\ i+j\geq 3}\frac{1}{n^{\lceil(i+j)/2\rceil}}C_{ij}\frac{\mu_y}{\mu_x}\frac{\partial^{i+j}}{(\partial\mu_x)^i(\partial\mu_y)^j}\frac{\mu_y}{\mu_x}.
\end{aligned}
\tag{24}
$$

Inserting Eqn. 22 and Eqn. 24 into Eqn. 16 results in

$$
\begin{aligned}
\mathrm{SE}_{\hat{\mathrm{f}}} = &\,2\frac{\mu_y^2}{\mu_x^2} + \frac{\mathrm{Var}(y)}{n\mu_x} + 3\,\mathrm{Var}(x)\frac{\mu_y^2}{n\mu_x^4} - 4\,\mathrm{Cov}(x, y)\frac{\mu_y}{n\mu_x^3} \\
&+ \sum_{i,j:\ i+j\geq 3}\frac{1}{n^{\lceil(i+j)/2\rceil}}C_{ij}\frac{\partial^{i+j}}{(\partial\mu_x)^i(\partial\mu_y)^j}\frac{\mu_y^2}{\mu_x^2} \\
&- 2\Bigg(\frac{\mu_y^2}{\mu_x^2} + \mathrm{Var}(x)\frac{\mu_y^2}{n\mu_x^4} - \mathrm{Cov}(x, y)\frac{\mu_y}{n\mu_x^3} \\
&+ \sum_{i,j:\ i+j\geq 3}\frac{1}{n^{\lceil(i+j)/2\rceil}}C_{ij}\frac{\mu_y}{\mu_x}\frac{\partial^{i+j}}{(\partial\mu_x)^i(\partial\mu_y)^j}\frac{\mu_y}{\mu_x}\Bigg) \\
= &\,\frac{1}{n}\left(\frac{\mathrm{Var}(y)}{\mu_x} + \mathrm{Var}(x)\frac{\mu_y^2}{\mu_x^4} - 2\,\mathrm{Cov}(x, y)\frac{\mu_y}{\mu_x^3}\right) \\
&+ \underbrace{\sum_{i,j:\ i+j\geq 3}\frac{1}{n^{\lceil(i+j)/2\rceil}}C_{ij}\left(\frac{\partial^{i+j}}{(\partial\mu_x)^i(\partial\mu_y)^j}\frac{\mu_y^2}{\mu_x^2} - \frac{2\mu_y}{\mu_x}\frac{\partial^{i+j}}{(\partial\mu_x)^i(\partial\mu_y)^j}\frac{\mu_y}{\mu_x}\right)}_{\in O\left(\frac{1}{n^2}\right)}.
\end{aligned}
\tag{25}
$$

Consequently, we may estimate $\mathrm{SE}_{\hat{\mathrm{f}}}$ via

$$
\widehat{\mathrm{SE}}_{\hat{\mathrm{f}}} := \frac{1}{n}\left(\frac{\hat{\mu}_y\hat{\sigma}_x^2}{\hat{\mu}_x^4} + \frac{\hat{\sigma}_y^2}{\hat{\mu}_x} - 2\frac{\hat{\mu}_y\hat{\sigma}_{xy}}{\hat{\mu}_x^3}\right),
\tag{26}
$$

which is consistent since the estimators $\hat{\mu}_y = \frac{1}{n}\sum_i y_i$, $\hat{\mu}_x = \frac{1}{n}\sum_i x_i$, $\hat{\sigma}_y^2 = \frac{1}{n-1}\sum_i (y_i - \hat{\mu}_y)^2$, $\hat{\sigma}_x^2 = \frac{1}{n-1}\sum_i (x_i - \hat{\mu}_x)^2$, and $\hat{\sigma}_{xy} = \frac{1}{n-1}\sum_i (x_i - \hat{\mu}_x)(y_i - \hat{\mu}_y)$ are consistent as well.

### B.3. Volume to Ratio Confidence Intervals

**Proposition 13 (Calibration-based Confidence Interval)** *Consider a segmentation model $g(z) = (g_A(z), g_B(z))$ with the random variable $z$ representing pixel inputs of instance $I$, and targets $y_A$ and $y_B$. On a validation (calibration) set $\mathcal{D}_{cal}$, define $q_{A,\delta/2}$ and $q_{B,\delta/2}$ as the $\frac{n+1}{n}(1-\frac{\delta}{2})$ quantile of the instance-wise volume bias or calibration errors of $g_A$ and $g_B$. Then, it holds with at least $1-\delta$ probability that*

$$\frac{\mathbb{E}\left[y_A \mid I\right]}{\mathbb{E}\left[y_B \mid I\right]} \in \left[\frac{\mathbb{E}\left[g_A(z) \mid I\right]}{\mathbb{E}\left[g_B(z) \mid I\right]} - \epsilon_{l,\delta}, \frac{\mathbb{E}\left[g_A(z) \mid I\right]}{\mathbb{E}\left[g_B(z) \mid I\right]} + \epsilon_{u,\delta}\right], \tag{27}$$

*where $\epsilon_{l,\delta} := \frac{\mathbb{E}[g_A(z)]}{\mathbb{E}[g_B(z)]} - \frac{\mathbb{E}[g_A(z)] - q_{A,\delta/2}}{\mathbb{E}[g_B(z)] + q_{B,\delta/2}}$, $\epsilon_{u,\delta} := \frac{\mathbb{E}[g_A(z)] + q_{A,\delta/2}}{\mathbb{E}[g_B(z)] - q_{B,\delta/2}} - \frac{\mathbb{E}[g_A(z)]}{\mathbb{E}[g_B(z)]}$ are the widths of the lower and upper calibration bounds, respectively.*

In experiments, CARE (V-Bias) takes the quantile of |V-Bias| (Popordanoska et al., 2021) as $q_{A,B}$ while CARE (ECE) considers ECE (Guo et al., 2017) quantiles. To combine both intervals, we make the following statement, which is analogous to multiple testing. This way, we can consider both uncertainties in practice.

Note that if $a \notin [b, c] \subseteq \mathbb{R}_{>0}$ then $\frac{1}{a} \notin \left[\frac{1}{c}, \frac{1}{b}\right]$ since $x \mapsto \frac{1}{x}$ is strictly negative monotone. We also make use of the subadditivity of probability measures (Resnick, 2003) given by

$$\mathbb{P}\left(\bigcup_i A_i\right) \le \sum_i \mathbb{P}(A_i). \tag{28}$$

This is also known as Boole's inequality. In the following, we denote the random variable $z$ as the pixel inputs of image instance $I$. As described in the main paper, $q_{A,\alpha}$ and $q_{B,\alpha}$ are empirically determined on a validation set as the $1-\alpha$ quantile of the image-wise calibration errors for $g_A$ and $g_B$. Then, for $\alpha \in [0,1]$ it holds that

$$
\begin{aligned}
\alpha = {} & \frac{\alpha}{2} + \frac{\alpha}{2} \\
\ge {} & \mathbb{P}\left(\mathrm{CE}_{A,I} \ge q_{A,\alpha/2}\right) + \mathbb{P}\left(\mathrm{CE}_{B,I} \ge q_{B,\alpha/2}\right) \\
\ge {} & \mathbb{P}\left(\left|\mathbb{E}\left[Y_A \mid I\right] - \mathbb{E}\left[g_A(z) \mid I\right]\right| \ge q_{A,\alpha/2}\right) + \mathbb{P}\left(\left|\mathbb{E}\left[Y_B \mid I\right] - \mathbb{E}\left[g_B(z) \mid I\right]\right| \ge q_{B,\alpha/2}\right) \\
\ge {} & \mathbb{P}\left(\left|\mathbb{E}\left[Y_A \mid I\right] - \mathbb{E}\left[g_A(z) \mid I\right]\right| \ge q_{A,\alpha/2} \vee \left|\mathbb{E}\left[Y_B \mid I\right] - \mathbb{E}\left[g_B(z) \mid I\right]\right| \ge q_{B,\alpha/2}\right) \\
= {} & \mathbb{P}\Big(\mathbb{E}\left[Y_A \mid I\right] \notin \left[\mathbb{E}\left[g_A(z) \mid I\right] - q_{A,\alpha}, \mathbb{E}\left[g_A(z) \mid I\right] + q_{A,\alpha}\right] \\
& \vee \mathbb{E}\left[Y_B \mid I\right] \notin \left[\mathbb{E}\left[g_B(z) \mid I\right] - q_{B,\alpha}, \mathbb{E}\left[g_B(z) \mid I\right] + q_{B,\alpha}\right]\Big) \\
= {} & \mathbb{P}\Big(\mathbb{E}\left[Y_A \mid I\right] \notin \left[\mathbb{E}\left[g_A(z) \mid I\right] - q_{A,\alpha}, \mathbb{E}\left[g_A(z) \mid I\right] + q_{A,\alpha}\right] \\
& \vee \frac{1}{\mathbb{E}\left[Y_B \mid I\right]} \notin \left[\frac{1}{\mathbb{E}\left[g_B(z) \mid I\right] + q_{B,\alpha}}, \frac{1}{\mathbb{E}\left[g_B(z) \mid I\right] - q_{B,\alpha}}\right]\Big) \\
\ge {} & \mathbb{P}\left(\frac{\mathbb{E}\left[Y_A \mid I\right]}{\mathbb{E}\left[Y_B \mid I\right]} \notin \left[\frac{\mathbb{E}\left[g_A(z) \mid I\right] - q_{A,\alpha}}{\mathbb{E}\left[g_B(z) \mid I\right] + q_{B,\alpha}}, \frac{\mathbb{E}\left[g_A(z) \mid I\right] + q_{A,\alpha}}{\mathbb{E}\left[g_B(z) \mid I\right] - q_{B,\alpha}}\right]\right).
\end{aligned}
\tag{29}
$$

It follows that for confidence level $1 - \alpha$ that

$$\frac{\mathbb{E}\left[Y_A \mid I\right]}{\mathbb{E}\left[Y_B \mid I\right]} \in \left[\frac{\mathbb{E}\left[g_A\left(z\right) \mid I\right] - q_{A,\alpha}}{\mathbb{E}\left[g_B\left(z\right) \mid I\right] + q_{B,\alpha}}, \frac{\mathbb{E}\left[g_A\left(z\right) \mid I\right] + q_{A,\alpha}}{\mathbb{E}\left[g_B\left(z\right) \mid I\right] - q_{B,\alpha}}\right] \tag{30}$$

Given the previous equation, it further holds that

$$
\begin{aligned}
\delta + \alpha &\geq \\
&\geq \mathbb{P}\left(\frac{\mathbb{E}\left[Y_A \mid I\right]}{\mathbb{E}\left[Y_B \mid I\right]} \notin \left[\frac{\mathbb{E}\left[g_A\left(z\right) \mid I\right]}{\mathbb{E}\left[g_B\left(z\right) \mid I\right]} - \epsilon_{l,\delta}, \frac{\mathbb{E}\left[g_A\left(z\right) \mid I\right]}{\mathbb{E}\left[g_B\left(z\right) \mid I\right]} + \epsilon_{u,\delta}\right]\right) \\
&\quad + \mathbb{P}\left(\frac{\mathbb{E}\left[g_A\left(z\right) \mid I\right]}{\mathbb{E}\left[g_B\left(z\right) \mid I\right]} \notin \left[\frac{\sum_i g_A\left(z_{i,I}\right)}{\sum_i g_B\left(z_{i,I}\right)} - \beta_{r,\alpha}, \frac{\sum_i g_A\left(z_{i,I}\right)}{\sum_i g_B\left(z_{i,I}\right)} + \beta_{r,\alpha}\right]\right) \\
&\geq \mathbb{P}\left(\frac{\mathbb{E}\left[Y_A \mid I\right]}{\mathbb{E}\left[Y_B \mid I\right]} \notin \left[\frac{\mathbb{E}\left[g_A\left(z\right) \mid I\right]}{\mathbb{E}\left[g_B\left(z\right) \mid I\right]} - \epsilon_{l,\delta}, \frac{\mathbb{E}\left[g_A\left(z\right) \mid I\right]}{\mathbb{E}\left[g_B\left(z\right) \mid I\right]} + \epsilon_{u,\delta}\right]\right. \\
&\qquad \left. \vee \frac{\mathbb{E}\left[g_A\left(z\right) \mid I\right]}{\mathbb{E}\left[g_B\left(z\right) \mid I\right]} \notin \left[\frac{\sum_i g_A\left(z_{i,I}\right)}{\sum_i g_B\left(z_{i,I}\right)} - \beta_{r,\alpha}, \frac{\sum_i g_A\left(z_{i,I}\right)}{\sum_i g_B\left(z_{i,I}\right)} + \beta_{r,\alpha}\right]\right) \\
&\geq \mathbb{P}\left(\frac{\mathbb{E}\left[Y_A \mid I\right]}{\mathbb{E}\left[Y_B \mid I\right]} \notin \left[\frac{\sum_i g_A\left(z_{i,I}\right)}{\sum_i g_B\left(z_{i,I}\right)} - \epsilon_{l,\delta} - \beta_{r,\alpha}, \frac{\sum_i g_A\left(z_{i,I}\right)}{\sum_i g_B\left(z_{i,I}\right)} + \epsilon_{u,\delta} + \beta_{r,\alpha}\right]\right).
\end{aligned}
\tag{31}
$$

From this follows that with at least probability $1 - \alpha - \delta$ that

$$\frac{\mathbb{E}\left[Y_A \mid I\right]}{\mathbb{E}\left[Y_B \mid I\right]} \in \left[\frac{\sum_i g_A\left(z_{i,I}\right)}{\sum_i g_B\left(z_{i,I}\right)} - \epsilon_{l,\delta} - \beta_{r,\alpha}, \frac{\sum_i g_A\left(z_{i,I}\right)}{\sum_i g_B\left(z_{i,I}\right)} + \epsilon_{u,\delta} + \beta_{r,\alpha}\right]. \tag{32}$$

### B.4. Debiased Ratio Estimation

The naive ratio estimator is biased due to the limited number of samples. Here we extend (Popordanoska et al., 2022) to derive a debiased ratio estimator to $\mathcal{O}(n^{-2})$. Firstly, the naive estimator is:

$$\hat{r} = \frac{\bar{y}}{\bar{x}} = \frac{\mu_y}{\mu_x}\left(\frac{\bar{y}}{\mu_y}\right)\left(\frac{\bar{x}}{\mu_x}\right)^{-1} = \frac{\mu_y}{\mu_x}\left(1 + \frac{\bar{y} - \mu_y}{\mu_y}\right)\left(1 + \frac{\bar{x} - \mu_x}{\mu_x}\right)^{-1}. \tag{33}$$

Then we expand $\left(1 + \frac{\bar{x} - \mu_x}{\mu_x}\right)^{-1}$ in Taylor series:

$$
\begin{aligned}
\hat{r} = \frac{\mu_y}{\mu_x}\Bigg(1 &+ \frac{(\bar{y} - \mu_y)}{\mu_y} - \frac{(\bar{x} - \mu_x)}{\mu_x} - \frac{(\bar{x} - \mu_x)(\bar{y} - \mu_y)}{\mu_y \mu_x} + \frac{(\bar{x} - \mu_x)^2}{\mu_x^2} \\
&+ \frac{(\bar{x} - \mu_x)^2(\bar{y} - \mu_y)}{\mu_x^2 \mu_y} - \frac{(\bar{x} - \mu_x)^3}{\mu_x^3} - \frac{(\bar{x} - \mu_x)^3(\bar{y} - \mu_y)}{\mu_x^3 \mu_y} + \frac{(\bar{x} - \mu_x)^4}{\mu_x^4}\Bigg) + \mathcal{O}(n^{-2.5})
\end{aligned}
\tag{34}
$$

The bias of $\hat{r}$ defined by $\mathbb{E}[\hat{r}] - r$ is written as:

$$\text{Bias}_r = \frac{\mu_y}{\mu_x}\left(\frac{1}{n}\left(\frac{\text{Var}(x)}{\mu_x^2} - \frac{\text{Cov}(x,y)}{\mu_x\mu_y}\right) + \frac{1}{n^2}\left(\frac{(\text{Cov}(x^2,y) - 2\mu_x\text{Cov}(x,y))}{\mu_x^2\mu_y}\right.\right. \tag{35}$$

$$\left.\left. - \frac{(\text{Cov}(x^2,x) - 2\mu_x\text{Var}(x))}{\mu_x^3} - \frac{3\text{Var}(x)\text{Cov}(x,y)}{\mu_x^3\mu_y} + \frac{3\text{Var}(x)^2}{\mu_x^4}\right)\right) \tag{36}$$

And a second-order debiased estimator is defined by $r_{corr,2} := \hat{r} - \text{Bias}_r$:

$$r_{corr,2} = \hat{r} - \frac{\mu_y}{\mu_x}\left(\frac{1}{n}\left(\frac{\text{Var}(x)}{\mu_x^2} - \frac{\text{Cov}(x,y)}{\mu_x\mu_y}\right) + \frac{1}{n^2}\left(\frac{(\text{Cov}(x^2,y) - 2\mu_x\text{Cov}(x,y))}{\mu_x^2\mu_y}\right.\right. \tag{37}$$

$$\left.\left. - \frac{(\text{Cov}(x^2,x) - 2\mu_x\text{Var}(x))}{\mu_x^3} - \frac{3\text{Var}(x)\text{Cov}(x,y)}{\mu_x^3\mu_y} + \frac{3\text{Var}(x)^2}{\mu_x^4}\right)\right) \tag{38}$$

Finally, we use plug-in estimators for empirical estimation:

$$\hat{r}_{corr,2} := \frac{\hat{\mu}_y}{\hat{\mu}_x}\left(1 - \frac{1}{n}\left(r_b^* - r_a^*\right) - \frac{1}{n^2}\left(\frac{(\widehat{\text{Cov}(x^2,y)} - 2\widehat{\mu}_x\widehat{\text{Cov}(x,y)})}{\widehat{\mu}_x^2\widehat{\mu}_y}\right.\right.$$
$$\left.\left. - \frac{(\widehat{\text{Cov}(x^2,x)} - 2\widehat{\mu}_x\widehat{\text{Var}(x)})}{\widehat{\mu}_x^3} - \frac{3\widehat{\text{Var}(x)}\widehat{\text{Cov}(x,y)}}{\widehat{\mu}_x^3\widehat{\mu}_y} + \frac{3\widehat{\text{Var}(x)}^2}{\widehat{\mu}_x^4}\right)\right) \tag{39}$$

$$r_a^* = \underbrace{\frac{\widehat{\text{Cov}(x,y)}}{\widehat{\mu}_x\widehat{\mu}_y}}_{=r_a}\left(1 + \frac{1}{(n-1)}\left(\frac{\widehat{\mu}_y\widehat{\text{Cov}(x^2,y)} + \widehat{\mu}_x\widehat{\text{Cov}(y^2,x)}}{\widehat{\text{Cov}(x,y)}\widehat{\mu}_x\widehat{\mu}_y} - 4\right)\right.$$
$$\left. - \frac{1}{(n-1)}\left(\frac{\widehat{\text{Var}(x)}}{\widehat{\mu}_x^2} + \frac{\widehat{\text{Var}(y)}}{\widehat{\mu}_y^2} + 2\frac{\widehat{\text{Cov}(x,y)}}{\widehat{\mu}_x\widehat{\mu}_y}\right)\right) \tag{40}$$

$$r_b^* = \underbrace{\frac{\widehat{\text{Var}(x)}}{\widehat{\mu}_x^2}}_{=r_b}\left(1 + \frac{4}{(n-1)}\left(\frac{\frac{1}{2}\widehat{\text{Cov}(x^2,x)}}{\widehat{\mu}_x\widehat{\text{Var}(x)}} - 1\right) - \frac{4}{(n-1)}\frac{\widehat{\text{Var}(x)}}{\widehat{\mu}_x^2}\right). \tag{41}$$

## Appendix C. Related Work

**Ratio-based biomarkers** are quantitative metrics that express the relative size, volume, or intensity of a target anatomical structure as a proportion of a reference region (Fig. 2). They are widely used across clinical domains to capture compositional, structural and functional changes, enabling standardized assessment of disease progression and treatment response. Examples include: ejection fraction – representing the fraction of blood ejected from the ventricle during each cardiac cycle; coronary artery stenosis – quantifying the percent narrowing of a coronary vessel, and fat fraction – measuring the proportion of fat

within an organ such as liver or kidney. Ratio-based biomarkers are particularly valuable for detailed tumor characterization. Key metrics include necrosis-to-tumor ratio (NTR) and core-to-tumor ratio (CTR), which quantify the internal structure of the tumor, as well as tumor invasion rate, which reflects the extent of tumor infiltration into surrounding tissues. In summary, the ratio-based measures offer standardized, comparable metrics that can be applied across imaging modalities, organs, and disease contexts.

Typically, clinicians compute these ratios using volumetric information from imaging data (*e.g.*, MRI) (Henker et al., 2019, 2017). With the advancement of computational pathology and the growing availability of annotated medical data, recent studies (Ye et al., 2023) have developed AI-based workflows for automated ratio assessment. These methods offer scalable and consistent evaluations, effectively overcoming the limitations of subjective human judgment in manual assessments. Despite promising developments, existing methods typically provide only point estimates (Ho et al., 2020), neglecting the associated uncertainty. Although intuitive, results computed from the outputs of segmentation networks inherit the known overconfidence tendency of neural networks (Guo et al., 2017). As a result, naïve ratio estimations from miscalibrated outputs are often biased from true values. Current research predominantly focuses on improving network calibration and segmentation accuracy (Rousseau et al., 2025; Wang et al., 2023; Mehrtash et al., 2020; Wang et al., 2022; Hatamizadeh et al., 2021), while overlooking the downstream task of biomarker estimation. Our work addresses this gap by proposing a confidence-aware framework for ratio estimation from segmentation models.

