# OpenReview forum: "CARE: Confidence-aware Ratio Estimation for Medical Biomarkers"
_MIDL.io/2026/Conference — MIDL 2026 Poster_

### Official Review · Reviewer_6GQk · 2025-12-19

**Confidence:** 4
**Preliminary Rating:** 5
**Final Rating:** 5

**Summary:**

Addresses uncertainty quantification for ratio-based biomarkers (necrosis-to-tumor ratio, tumor-to-kidney ratio) derived from segmentation outputs. Decomposes uncertainty into estimation-based variance and calibration-based error. Uses Markov inequality bounds with conformal prediction to provide adaptive confidence intervals without additional training. Key finding: miscalibration is the dominant source of uncertainty. Experiments on brain tumor MRI (MSD-Task01, BraTS21) and kidney CT (KiTS23) show tighter, more adaptive intervals than standard conformal prediction baselines.

**Strengths:**

- Clinical problem is well-motivated -- point estimates insufficient for high-stakes treatment decisions
- Theoretical framework combining Markov bounds with conformal prediction avoids strong distributional assumptions
- Key finding that miscalibration dominates over statistical variance (Figure 4c) directs future research toward better calibration
- Model-agnostic: works across nnUNet, nnFormer, UNETR++ and three datasets
- Confidence intervals adapt to task difficulty (wider for small tumors), unlike CQR which cannot distinguish difficulty levels

**Weaknesses:**

- No computational cost analysis -- claims "minimum assumptions" and "single forward pass" but no timing data for clinical deployment assessment
- Calibration set size not investigated -- uses 10% holdout but doesn't establish minimum validation set size for stable quantile estimation, critical for small datasets
- Domain shift acknowledged in Section 5 limitations but not empirically tested -- if test scanner differs from validation, calibrated quantiles become invalid
- Grid search for \\(\\alpha\\) and \\(\\delta\\) lacks sensitivity analysis -- unclear how robust the method is to different choices
- No clinical validation or comparison to radiologist uncertainty practices
- Taylor expansion (Proposition 9) assumes i.i.d. pixels. Paper acknowledges this is violated by spatial correlation (Section 3) and claims Markov bounds compensate, but doesn't quantify how much looser bounds become

**Detailed Comments:**

- Table 1: Add significance markers showing which methods differ from CARE (eg ACQR)
- Taylor expansion (Proposition 9) assumes i.i.d. pixels. Paper acknowledges spatial correlation violates this and argues Markov bounds compensate (Appendix B.2), but doesn't quantify the effect on the SE estimator

**Justification Of Final Rating:**

The rebuttal adequately addresses the computational cost, grid search sensitivity, and theoretical concerns. The core contribution (decomposing ratio uncertainty, identifying miscalibration as dominant, providing adaptive intervals) remains sound.

**Justification Of The Preliminary Rating:**

Addresses an important gap in trustworthy medical AI with a principled theoretical framework. The combination of Markov bounds with conformal prediction is well-motivated, and the finding that miscalibration dominates is actionable. Experimental validation across 3 datasets and 4 architectures is thorough. Weaknesses (computational analysis, calibration set size sensitivity, domain shift testing) are addressable and don't undermine the core contribution.

**Questions To Address In The Rebuttal:**

1. How much inference time does computing CARE intervals add vs. base segmentation?
2. Show coverage/interval width vs. calibration set size (5%, 10%, 20%, 30%)
3. What interval widths are clinically actionable? When too wide for decision-making?
4. Can CARE readily extend to biomarkers involving ratios of three or more tissue types (e.g., necrotic vs. enhancing vs. healthy)? May be relevant to other fields if so.

---

> ### Author Response · Authors · 2026-01-24
> **Response to 6GQk**
>
> > **Inference time**
>
> We appreciate the suggestion of reporting inference time. In **Tab.4**, our method only adds around 5 ms after segmentation, which shows great efficiency.
>
> > **Data size and domain shift**
>
> Please refer to the general response and the response to r7D1.
>
> > **Concerns about grid search**
>
> We provide a comparison with “fixed $(\alpha, \delta)$” in **Tab.5**, which shows the advantage of grid search.
>
> > **Clinical action**
>
> As the use case in Fig.1b shows, we don’t directly give treatment recommendations, but provide confidence intervals for the estimated ratio. According to the user-defined threshold (e.g. 0.25), predictions whose confidence interval goes across the threshold (e.g. >0.25) are not reliable and trigger a human check.
>
> > **Violation of i.i.d**
>
> The Markov’s bounds have relatively small contributions to the overall uncertainty (Fig.4c). Therefore, the tightness is mostly determined by the miscalibration of models. Please also refer to the response to r7D1.
>
> > **Multiple tissue types**
>
> In the necrosis-to-tumor ratio (NTR), the denominator already includes many tissue types (the whole tumor=necrosis+edema+enhancing tumor). If the multiple tissues here mean the proportion among more tissue types (e.g. A:B:C), CARE is able to extend to this task by formulating “necrotic vs. enhancing vs. healthy” as the numerator with a shared denominator x (e.g. the whole organ).

---

> > ### Comment · Reviewer_6GQk · 2026-01-25
> > **The authors addressed my main concerns**
> >
> > 1. **Inference time**: 5ms overhead (Table 4) is negligible for clinical deployment. Concern resolved.
> >
> > 2. **Grid search sensitivity**: Table 5 comparison to fixed parameters demonstrates the value of adaptive selection. Satisfactory.
> >
> > 3. **Clinical workflow**: The threshold-crossing paradigm (flag for human review rather than direct treatment recommendation) is a sensible clinical integration strategy.
> >
> > 4. **i.i.d. assumption**: The argument that Markov bounds contribute minimally to overall uncertainty (Figure 4c) is reasonable -- if miscalibration dominates, tightness of the statistical bounds matters less.
> >
> > 5. **Multiple tissue types**: Extension via shared denominator is straightforward and addresses the extensibility question.
> >
> >
> > ## Final Rating
> > 5: Accept
> >
> > ## Final Rating Justification
> > The rebuttal adequately addresses the computational cost, grid search sensitivity, and theoretical concerns. The core contribution (decomposing ratio uncertainty, identifying miscalibration as dominant, providing adaptive intervals) remains sound.

---

### Official Review · Reviewer_FmgD · 2026-01-09

**Confidence:** 3
**Preliminary Rating:** 3
**Final Rating:** 4

**Summary:**

In this work, the authors study the estimation of the uncertainty of ratio-based biomarkers obtained from automatic segmentations, such as the necrosis-to-tumor ratio and the tumor-to-kidney ratio. Their confidence interval estimation method takes into account both the inherent confidence interval from the bias and the variance of the task and the uncertainty caused by miscalibration, which typically occur in deep learning based methods.

**Strengths:**

- The problem studied by this work is very clinically relevant. While most research works focus on improving the segmentation performance in all possible contexts, the authors focus on the next step, which is often to extract biomarkers to assess disease progression or guide decisions.
- The paper, in particular sections 2 and 3, is well explained and rather easy to follow considering its complexity.
- The proposed method allows for capturing the uncertainty of both estimation-based error and calibration-based error, which are two common sources of uncertainty of deep learning models.
- The method is carefully validated on two large datasets with nested cross-validation, and several experiments have been conducted to analyze the results and evaluate the robustness of the method (ablation on u(x), effect of temperature, etc.)

**Weaknesses:**

- The main body of the paper lacks a proper Related Works section. I hope that the two additional pages allowed for the rebuttal phase will allow the authors to include the section they put at the end of the appendices in the main body, for its completeness.
- I feel like the experiment setup and results could be more thoughtfully explained to ensure that the readers fully understand the experiments and the results. In particular, the parts on the Coverage guarantee and Adaptiveness should be clearer (please refer to the Detailed Comments section)

**Detailed Comments:**

- For the UQ baselines, I don't understand why having chosen the "3sigma" confidence intervals, while C=0.68 is the confidence level for a deviation of 1sigma for a normal distribution. Could you please explain this choice.
- I am not sure to understand what the coverage rate is. I think the authors should add a more formal definition of the metric. In particular, for me, the sentence "We report coverage rate (%) of different UQ methods at 0.68 confidence level in Table 1, which measures the proportion of samples whose true values fall within the confidence intervals." suggests that the ideal coverage rate for C=0.68 should be 68%. If that is the case, then wouldn't it mean that the best method in Table 1 is CQR?
- Similarly, I think that the paragraph in the results section about the Adaptiveness should be worded more clearly. More specifically, In Figure 3.a), I don't understand why CARE interval distributions are more desirable than ACQR distributions, and why it is relevant to show a better adaptiveness at the sample level of the proposed method
-  Although I understand that the uncertainty should be correlated to the difficulty to segment the tumors. I intuitively see that between the categories S and M,  and S and L , the difficulty of segmentation might be quite different but it could not be the case between M and L. I think that reporting the segmentation performance on the three subgroups S, M and L could help better interpreting the Figure 3.b). Morevover, I think that in the sentence "widths are proportional to the segmentation difficulty", the term "proportional" is a bit misused.
- In Figure 4.a, I don't understand what the red dashed line adds to the graph ? Isn't the confidence already given by the x-axis ? On this figure as well and same question as my in my comment above about Table 1, isn't the coverage suppoes to match the confidence (i.e. the red dashed line). If that is the case then CQR is a better method, for this metric, than ACQR and the proposed CARE.
- [minor] I think that in Table 1, the light green background colour of the row "CARE (V-Bias)" should be removed, as it biases the perception and analysis of the metrics. In my opinion, the authors should opt for a more neutral approach, such as mentioning "Ours" in the first column, and put the best metrics for each column in bold if they want to.

**Justification Of Final Rating:**

I would like to thank the authors for their responses to my comments and those of the other reviewers. The rebuttal phase has improved the quality of an already good paper and reinforced my belief in the value of the authors' work for the field of medical imaging. For these reasons, I am raising my rating to weak accept.

**Justification Of The Preliminary Rating:**

The paper is overall well written and techinically sound. The proposed method is interesting and tackle a clinically relevant problem. However, I think that the results section should be more precise and provide more details about the experiments. For now, my rating is borderline as I am not fully sure to understand the benefits of the proposed method, but I would be inclined to raise my rating if the authors answer my main concerns.

**Questions To Address In The Rebuttal:**

The authors are encouraged to respond to the weaknesses and the detailed comments.

---

> ### Author Response · Authors · 2026-01-24
> **Response to FmgD**
>
> > **Related work**
>
> We acknowledge that ratio-based biomarkers are still an underexplored topic, where a clinical background is necessary to bridge understanding. For completeness, we move parts of "related work" into the main paper (Sec.5).
>
> > **Metric definitions**
>
> We acknowledge the definition of coverage rate in Sec.4.2 is not clear. Following the reviewer’s advice, we added a formal definition in **Def.3 (Sec.2)**. We would like to clarify that CQR is non-adaptive, and a qualified method should achieve "at least" not "just" 68% for coverage rate, with **higher rates indicating superior performance**.  Therefore, we evaluate UQ methods using two metrics jointly: (1) we first identify methods that satisfy the desired coverage level (Tab.1), and then compare the adaptiveness and tightness among the qualified methods (Fig.3).
>
> > **Clarification on Fig.4a**
>
> The dashed line is used to aid understanding, i.e. methods above it achieve the confidence level. We acknowledge that a method covering just 68% yields the narrowest bounds, but due to safety concerns in clinical settings, the borderline method (e.g. CQR) is unreliable and readily collapses. In addition, CQR is non-adaptive by design. We revised the writing in Sec.5.1, explaining how to read Tab.1 and Fig.3 jointly. Both ACQR and CARE have coverage guarantee and adaptiveness, but CARE flags fewer samples for human-check.
>
> > **Clarification on Fig.3a**
>
> Stratified into 3 groups, ACQR concentrates in the middle region (around 0.45), which is not desired to be a really "informative and adaptive" uncertainty measure. In contrast, ours yields clearly distinguishable intervals in narrow (trustworthy) or wide (unreliable) ranges, which easily supports a human-check alarm.
>
> > **Clarification on Tab.1**
>
> C=0.68 and the 1sigma range are widely adopted values for confidence intervals. Yet, the 1sigma ranges of ensemble and dropout are too narrow to cover the confidence level, even "3sigma" settings fail to provide useful intervals, which further proves the weakness of Bayesian methods. For understanding, we also add 1sigma results in Tab.1.
>
> > **Advice on presentations**
>
> Thanks for your advice on the Tab.1 format. We use a green background to highlight four methods that satisfy C=0.68. We also appreciate the advice for *"proportional"* in page 9, where we’ve changed to "associated".
>
>
> > **Segmentation metrics**
>
> To get a comprehensive understanding, we report IoU and Dice scores in **Tab.3**. We acknowledge that group M and L don’t differ a lot in segmentation performance, while the ratio estimation is more complex for considering both tumor and necrosis. From the overall tendency, the performance degradation of tumor segmentation (Tab.3) and ratio estimation (MSE in Fig.3b) indicate a relationship between tumor size and prediction difficulty. Therefore, our adaptiveness on the interval width is justified.

---

> > ### Comment · Reviewer_FmgD · 2026-02-01
> >
> > Thank you for your answers to my concerns and comments. As said in my official review, I feel now more confident to recommend the acceptance of your work, and I have then changed my rating to weak accept.

---

### Official Review · Reviewer_r7D1 · 2026-01-09

**Confidence:** 4
**Preliminary Rating:** 3
**Final Rating:** 4

**Summary:**

This paper addresses the lack of uncertainty measures in ratio-based biomarkers in clinical practice by proposing CARE (a unified Confidence-Aware Ratio Estimation) for Medical Biomarkers to generate trustworthy Confidence Intervals (CI) without retraining the segmentation model. The framework performs uncertainty analysis in 2 ways by identifying sources of error and quantifying their overall impact on the CI : (i) addressing the issue of statistical bias and variance in the ratios by creating a ratio estimator bound using Markov's inequality based on the variance of the numerator and denominator ; (ii) segmentation networks have a calibration error (i.e. mostly overconfident). The paper proves that Volume Bias is mathematically associated with the Calibration Error and derive a squared estimator error from volume predictions. Then the work applies Conformal Prediction to produce statistically sound confidence intervals/miscalibration bounds.

**Strengths:**

1. This paper is well-written, well-structured and offers a novel framework for uncertainty measurement in RBBs thereby addressing a pressing clinical problem.

2. The paper empirically proves that Calibration Error from Volume predictions is a major source of error (vs simple statistical variance).

3. It proposes a plug and play framework and works with any segmentation model without requiring additional retraining.

4. CARE consistently hits the target coverage versus Bayesian methods which are under-covered.

5. The CARE framework is flexible - it creates wider intervals for small/hard tumors and tighter intervals for large/easy ones allowing for clinical flexibility.

6. The method is computationally efficient - it works in a single forward pass compared to ensembles which require multiple passes.

**Weaknesses:**

1. Context for large bounds for small tumors: Markov's inequality is known for being extremely conservative. The intervals for small tumors may be too wide to be clinically useful.

2. The proposed framework requires validation data to tune the confidence intervals. It assumes that the test patient data comes from the same distribution as the validation set. However, this may not be clinically realistic given that domain shift is an important issue in medical imaging. Moreover, conformal prediction guarantees tend to collapse under domain shifts.

3. Going by Table 1, there does not seem to be much of an improvement in the coverage compared to ACQR across all 4 segmentation models.

4. Figure 4: Confidence level is similar to ACQR. For (c) Uncertainty ablation, comparison with ACQR is not shown.

5. Is there a quantifiable figure on the number of patients experimentally flagged for Clinician Review using CARE ? Example: If the figure is close to 80% then it makes the framework less helpful versus a more practical figure that eases clinical burden.

**Detailed Comments:**

No further comments other than the ones given in the previous sections.

**Justification Of Final Rating:**

The authors have satisfactorily addressed the review comments in the rebuttal and made changes to the paper that make it suitable for publication to the MIDL conference. After careful deliberation, I have decided to increase my original rating to 4.

**Justification Of The Preliminary Rating:**

The proposed framework is definitely clinically important for improving trustworthiness and clinical safety by providing uncertainty estimates for RBBs. However, a few questions remain regarding the methodology and the results which need to be addressed. I am going with Borderline for now, and I am to open to revising my score based on the author feedback.

**Questions To Address In The Rebuttal:**

1. Given the issues raised regarding Markov's inequality, how would tighter bounds like Chebyshev's inequality or Bernstein bounds perform in terms of providing a tighter CI without compromising on clinical safety ?

2. Does the CARE framework address the issues of domain and distribution shift in clinical data ?

3. Table 1 - CARE performs as well as ACQR. Are there any additional benefits meant to be shown in this table that have not been shown.

4. Please refer to the comments in the "Weaknesses" section regarding Figure 4 and address them.

---

> ### Author Response · Authors · 2026-01-24
> **Response to r7D1**
>
> > **Markov's inequality**
>
> We appreciate this insightful comment on Markov's inequality. We chose it because:
>
> - Empirically, the overall confidence interval is dominated by the network’s miscalibration (Fig.4c). Therefore, we introduce tight CARE (V-Bias) as the default method and conservative CARE (ECE) as the upper bound. Notably, small tumors have wide bounds since they are treated as difficult samples. In comparison, the bounds for large tumors (simple samples) are very narrow.
>
> - Markov’s inequality requires only the first-order moment of the distribution and does not assume knowledge of the variance. Given the difficulty in estimating higher-order moments (like variance), using Markov's Inequality ensures a sound and distribution-agnostic bound.
>
> > **Domain-shift concerns**
>
> - Please refer to “General response”.
>
> > **Clarifications on Tab.1 and Fig.4**
>
> - Tab.1: We agree that ACQR achieves a comparable coverage rate to ours. However, the coverage rate needs to be combined with the adaptiveness to assess a UQ method, where ACQR is too conservative to provide informative intervals (Fig.3b). From method design, ACQR gets the coverage rate by adopting the max tumor size (Remark 4) as the prior knowledge, which is less feasible in practice. In comparison, CARE gets a good coverage rate and maintains adaptiveness from two metrics, but assumes no prior from method design.
>
> - Fig.4: Fig.4a extends Tab.1 to more confidence levels, and indicates that CARE and ACQR have coverage guarantee. Yet, CARE doesn’t assume the prior of tumor size. Moreover, to avoid the overlap with Fig.3b, we show the **ablation of CARE** in Fig.4c, to indicate the uncertainty contribution from estimation and miscalibration.  In the camera-ready version, we will split Fig.4 into (a) general comparison, and (b-c) ablation study for a clear understanding.
>
> - Quantifiable figure: We’re still at the methodology stage, and the clinical justification is our next stage. Yet, ours is tighter than ACQR and is expected to flag fewer samples than ACQR.

---

### Author Rebuttal · Authors · 2026-01-24

**Rebuttal:**

We thank all reviewers for their thoughtful and constructive feedback.

Reviewers consistently emphasized the *"clinical significance"* and *"key finding of uncertainty sources"*. Reviewer **r7D1** noted that *"it’s flexible for small/large tumors and computationally efficient"*, and Reviewer **FmgD** highlighted *"well explained and easy to follow"*. Finally, Reviewer **6GQk** appreciates that *"the confidence intervals adapt to task difficulty"*.

Reviewers also raised several thoughtful questions regarding our paper. While we provide detailed responses to each reviewer below, we first address the common concern regarding domain shifts:

(1) **Methodological Foundation**: We prioritize establishing a methodological foundation with theoretical guarantees in the i.i.d. setting. This serves as a prerequisite for future extensions to distribution-shift scenarios using label-free estimators [1, 2].

(2) **Theoretical Consistency**: The assumption of identical distribution between the validation and test sets is a foundational premise for most prediction and calibration frameworks [3], including Conformal Prediction (CP), which our work is built on.

(3) **Clinical Feasibility**: A local validation set is often readily available and manageable within a hospital's private infrastructure [4]. Empirically, CARE demonstrates remarkable robustness even with limited data; as shown in Table 2, our method maintains performance with just 10 samples, where distribution assumptions are harder to strictly satisfy.

*Reference:*

[1] Penso, C., et al. Calibration of Network Confidence for Unsupervised Domain Adaptation Using Estimated Accuracy. In European Conference on Computer Vision (ECCV), 2024.

[2] Popordanoska, M., et al. LaSCal: Label-shift calibration without target labels. In Advances in Neural Information Processing Systems (NeurIPS), 2024.

[3] Vapnik, V. N. An overview of statistical learning theory. IEEE Transactions on Neural Networks, 10(5):988–999, 1999.

[4] Sheller, M. J., et al. Federated learning in medicine: Facilitating multi-institutional collaborations without sharing patient data. Scientific Reports, 10(1):12598, 2020.

**Supporting Material:**

/attachment/785cb26fb8debd9c096807b286ef92593b7d4930.pdf

---

### Meta-Review · Area_Chair_8SZy · 2026-02-09

**Recommendation:** Accept (Poster)
**Confidence:** 5

**Metareview:**

This paper addresses the lack of uncertainty measures in ratio-based biomarkers in clinical practice by proposing CARE (a unified Confidence-Aware Ratio Estimation) for Medical Biomarkers to generate trustworthy Confidence Intervals (CI) without retraining the segmentation model.
It decomposes uncertainty into estimation-based variance and calibration-based error and uses Markov inequality bounds with conformal prediction to provide adaptive confidence intervals without additional training.
Reviewers first raised some concerns regarding the methodological foundation, the theoretical consistency and the clinical feasibility.
The authors have satisfactorily addressed the review comments in the rebuttal. All reviewers agreed the paper is suitable for publication to the MIDL conference.

---

### Decision · Program_Chairs · 2026-02-13

Accept (Poster)